# Linker histone variant H1.2 is a brake on white adipose tissue browning

Yangmian Yuan[1], Yu Fan[1], Yihao Zhou[1], Rong Qiu[1], Wei Kang[1], Yu Liu[1], Yuchen Chen[2], Chenyu Wang[1], Jiajian Shi[2], Chengyu Liu[3], Yangkai Li[4], Min Wu[1], Kun Huang[2], Yong Liu[1] & Ling Zheng[1] ✉

Adipose-tissue is a central metabolic organ for whole-body energy homeostasis. Here, we find that highly expressed H1.2, a linker histone variant, senses thermogenic stimuli in beige and brown adipocytes. Adipocyte H1.2 regulates thermogenic genes in inguinal white-adipose-tissue (iWAT) and affects energy expenditure. Adipocyte H1.2 deletion (H1.2AKO) male mice show promoted iWAT browning and improved cold tolerance; while over-expressing H1.2 shows opposite effects. Mechanistically, H1.2 binds to the promoter of *Il10rα*, which encodes an Il10 receptor, and positively regulates its expression to suppress thermogenesis in a beige cell autonomous manner. Il10rα overexpression in iWAT negates cold-enhanced browning of H1.2AKO male mice. Increased H1.2 level is also found in WAT of obese humans and male mice. H1.2AKO male mice show alleviated fat accumulation and glucose intolerance in long-term normal chow-fed and high fat diet-fed conditions; while Il10rα overexpression abolishes these effects. Here, we show a metabolic function of H1.2-Il10rα axis in iWAT.

Fat tissue is a highly plastic and dynamic organ, including white adipose tissue (WAT) and brown adipose tissue (BAT), which are responsible for energy storage and dissipation, respectively[1]. Dysfunctions of adipose tissue are closely associated with increased risk of metabolic diseases such as obesity and type 2 diabetes[2]. Adipocyte is the principal cell type in fat tissue. Adipocyte has two major forms, white adipocyte and brown adipocyte; and under cold exposure or β3-adrenergic stimulation, a third form, beige adipocyte with a brown-adipocyte-like thermogenic phenotype, is induced in WAT in a process termed browning[3]. Inducing the number of thermogenic brown and beige adipocytes activates energy expenditure, prevents high-fat diet (HFD) induced obesity, ameliorates hyperlipidemia, and improves systemic glucose homeostasis and insulin sensitivity in mice[4,5]. Thus, understanding the mechanisms controlling browning shall help to develop new therapies against obesity and associated metabolic complications.

Besides mature adipocytes, there are adipocyte precursor cells, endothelial cells, neural cells, and immune cells that reside within fat tissue[6]. Immune cells play important roles in adipose tissue inflammation and the resultant metabolic dysfunctions[2]. Recently, pronounced effects of immune cell-adipocyte crosstalk on the expansion and activation of beige adipose tissue have been suggested[7]. Il10, an anti-inflammatory cytokine secreted by immune cells including macrophages, B cells, and T cells[8], inhibits adrenergic signaling and adipose thermogenesis through its receptor Il10rα (Il10 receptor subunit alpha) expressed in mature adipocytes[9,10], indicating that thermogenesis is regulated by signaling network between immune cells and adipocytes.

[1]Hubei Key Laboratory of Cell Homeostasis, Frontier Science Center for Immunology and Metabolism, College of Life Sciences, Wuhan University, 430072 Wuhan, China. [2]School of Pharmacy, Tongji Medical College and State Key Laboratory for Diagnosis and Treatment of Severe Zoonotic Infectious Diseases, Huazhong University of Science and Technology, 430030 Wuhan, China. [3]Department of Transfusion Medicine, Wuhan Hospital of Traditional Chinese and Western Medicine, Tongji Medical College, Huazhong University of Science and Technology, 430030 Wuhan, China. [4]Department of Thoracic Surgery, Tongji Hospital, Tongji Medical College, Huazhong University of Science and Technology, 430030 Wuhan, China. ✉e-mail: lzheng@whu.edu.cn

Many epigenetic factors are involved in the regulation of thermogenesis, including HDAC3, UTX, EHMT1, and TET1[11–15]. However, as key regulators of chromatin structure and dynamics[16], the roles of histone variants in adipose tissue remain unclear. Previously, the critical role of macroH2A1.1 in white adipocyte differentiation has been reported[17]; and overexpression of a mutant of histone variant H3.3 (H3.3K4M) in BAT precursor cells has been shown to impair BAT development in mouse[18], indicating that histone variants may involve in the regulation of metabolic homeostasis.

From *Caenorhabditis elegans* to humans, H1.2 is the most conserved variant among all five somatic histone H1 variants (H1.1-H1.5)[19], implicating its evolutional and functional significance. H1.2 plays important functions in cellular processes like apoptosis, autophagy, and DNA damage[20–22]; and is associated with tumorigenesis, including lymphoma[23], hepatic carcinoma[24], and bladder cancer[25]. However, the function of H1.2 in metabolism, especially in adipose tissues, remains unknown.

Here, we find that in mature adipocytes, highly expressed H1.2 senses thermogenic stimuli in iWAT and BAT. Adipocyte-specific H1.2 knockout (H1.2AKO) mice show enhanced iWAT browning and energy expenditure under cold stimulation; while overexpressing H1.2 shows opposite effects. H1.2AKO mice also display improved metabolic status under long-term normal chow (NC)-fed and HFD-fed conditions. Overexpression of Il10rα in iWAT of H1.2AKO mice inhibits cold-induced browning, and alleviates the metabolic beneficial effect under long-term NC-fed or HFD-fed conditions. Thus, metabolic roles of H1.2-Il10rα axis in beige adipocytes are suggested.

## Results

### Adipocyte H1.2 senses temperature and adrenergic stimulations

Tissue expression profile of H1.2 was analyzed using adult C57BL/6 male mice. Among the tissues examined, the highest expression level of H1.2 was found in BAT and iWAT, while medium expression level was found in lung and heart, and low level in other tissues including eWAT (Fig. 1a). Meanwhile, the transcriptional level of *H1.2* in adipose tissues was in the rank order of BAT, iWAT, and eWAT (Fig. 1b). Moreover, the transcriptional levels of other somatic H1 variants were extremely low in both iWAT and BAT of mouse (Supplementary Fig. 1a, b), while higher *H1.2* level was also found in subcutaneous fat of human subjects (Supplementary Fig. 1c). Considering that iWAT and BAT are involved in adaptive thermogenesis, whether H1.2 senses stimulations of cold exposure or $\beta_3$-adrenergic agonist was investigated. As expected, Ucp1 (uncoupling protein 1, a thermogenesis marker) was induced by cold exposure (6 °C for 3 days) or CL316,243 (a $\beta_3$-adrenergic agonist) treatment in both iWAT and BAT; meanwhile, a significant increase of H1.2 was also detected (Fig. 1c, d and Supplementary Fig. 2a–c). Consistently, Ucp1 and H1.2 were downregulated in iWAT and BAT under thermoneutral environment (30 °C) in which the adaptive thermogenesis was inhibited (Fig. 1e, f).

To compare H1.2 level in different cell types of adipose tissue, the stromal vascular fraction (SVF) and mature adipocytes were successfully isolated, with *Cd45* as the marker for SVF[26] and *Adipoq/Plin1* as the markers for mature adipocytes[27] (Supplementary Fig. 2d, e). In iWAT and BAT, H1.2 was preferentially expressed in mature adipocytes rather than in SVF (Fig. 1g, h). Consistently, low level of H1.2 was found in primary preadipocytes derived from iWAT or BAT at Day 0; while H1.2 was gradually increased during beige or brown adipocytes differentiation (Fig. 1i–l). Collectively, these data suggest that H1.2 may involve in adaptive thermogenesis in mature beige and brown adipocytes.

### Adipocyte-specific H1.2 deletion promotes energy expenditure

Since H1.2 is highly expressed in mature adipocytes (Fig. 1g, h), we generated adipocyte-specific H1.2 knockout mice (H1.2AKO; Supplementary Fig. 3a, b) to explore whether this temperature-responsive

H1.2 alteration regulates adaptive thermogenesis. Significantly reduced H1.2 in eWAT, iWAT, and BAT, but not in liver and skeletal muscle, was found in H1.2AKO mice (Supplementary Fig. 3c–f). Immunofluorescent staining also confirmed the efficient and selective knockout of adipocyte H1.2 (Supplementary Fig. 3g). Meanwhile, transcriptional levels of the other somatic H1 variants were not affected in iWAT and BAT of H1.2AKO mice (Supplementary Fig. 3h, i).

Under NC-feeding, young (10-week-old) H1.2AKO mice showed no difference in body weight, fat mass, or lean mass compared with those of wildtype (WT) littermates (Fig. 2a, b). Moreover, anatomical results showed no difference in weights of eWAT/iWAT/BAT and liver, as well as in adipocyte size of iWAT/eWAT/BAT, between H1.2AKO mice and their WT littermates (Fig. 2c and Supplementary Fig. 4). Thus, we performed RNA sequencing (RNA-seq) to unbiasedly investigate altered genes or pathways in iWAT and BAT of H1.2AKO mice. Deficiency of H1.2 in iWAT and BAT led to 731 and 987 differentially expressed genes, among which, 201 genes were found in both tissues (Fig. 2d). Further analysis of these commonly altered genes showed a strong enrichment for PPARα signaling pathway (Fig. 2e), indicating that H1.2 in adipocyte may regulate energy catabolism. Therefore, $O_2$ consumption and $CO_2$ production were monitored with metabolic cages to calculate energy expenditure. Elevated $O_2$ consumption and energy expenditure, especially during light (resting) phase, were observed in NC-fed young H1.2AKO mice, indicating that H1.2 in adipocyte may upregulate basal metabolic rates (Fig. 2f, g). Consistently, significantly upregulated Ucp1 was found in iWAT of H1.2AKO mice (Fig. 2h, i).

### H1.2 in adipocyte regulates cold-induced iWAT browning

Upregulated Ucp1 in iWAT of NC-fed young H1.2AKO mice inspired us to investigate the role of adipocyte H1.2 in cold-induced browning. Housing young H1.2AKO and WT mice at 6 °C for 3 days, higher core body temperature, increased $O_2$ consumption, and energy expenditure were found in H1.2AKO mice (Fig. 3a–c). Histological examination revealed more multilocular adipocytes (demonstrated by H&E staining) with increased Ucp1 staining in iWAT of cold-exposed H1.2AKO mice (Fig. 3d). Consistently, qPCR revealed upregulated thermogenic genes such as *Ucp1*, *Dio2* (deiodinase type 2), and *Cidea* (cell death inducing DFFA like effector a), as well as lipid β-oxidation related genes like *Cpt1b* (carnitine palmitoyltransferase 1b), *Acot1* (acyl-CoA thioesterase 1) and *Acsl1* (acyl-CoA synthetase long-chain family member 1) in iWAT of cold-challenged H1.2AKO mice (Fig. 3e). Meanwhile, transcriptional levels of other somatic H1 variants were not altered in iWAT of cold-challenged H1.2AKO mice (Supplementary Fig. 5a). In addition to Ucp1-mediated canonical thermogenesis, Ucp1-independent thermogenic mechanisms, such as creatine and calcium cycling signaling, are also involved in beige adipocyte thermogenesis[28,29]. Thus, mRNA levels of *Gatm* (glycine amidinotransferase), *Gamt* (guanidinoacetate *N*-methyltransferase) and *Ckmt1* (mitochondrial creatine kinase 1) (key genes involved in creatine signaling-mediated thermogenesis[28]), and *Serca2b* (cardiac sarcoplasmic reticulum $Ca^{2+}$-ATPase, a key gene involved in calcium cycling-mediated thermogenesis[29]), were examined in iWAT of cold-challenged H1.2AKO mice with no change observed (Supplementary Fig. 5b), indicating that the enhanced thermogenesis in beige adipose tissue of H1.2AKO mice predominantly rely on Ucp1. However, no significant change in adipocyte morphology, Ucp1 expression, or transcriptional levels of thermogenic genes and other somatic H1 variants was found in BAT of cold-challenged H1.2AKO mice (Supplementary Fig. 6). Together, these data indicated H1.2 as a negative regulator of adaptive thermogenesis through acting on beige adipocyte in iWAT, rather than in BAT, under cold exposure.

To further verify its role in cold-induced iWAT browning, H1.2 was overexpressed by in situ injection of adeno-associated virus-packed mouse H1.2 (AAV-H1.2) into iWAT pads of both sides. Successful H1.2 overexpression in iWAT was achieved at 3 weeks after the injection (H1.2OE mice; Fig. 3f, g). H1.2 overexpression in iWAT did not affect

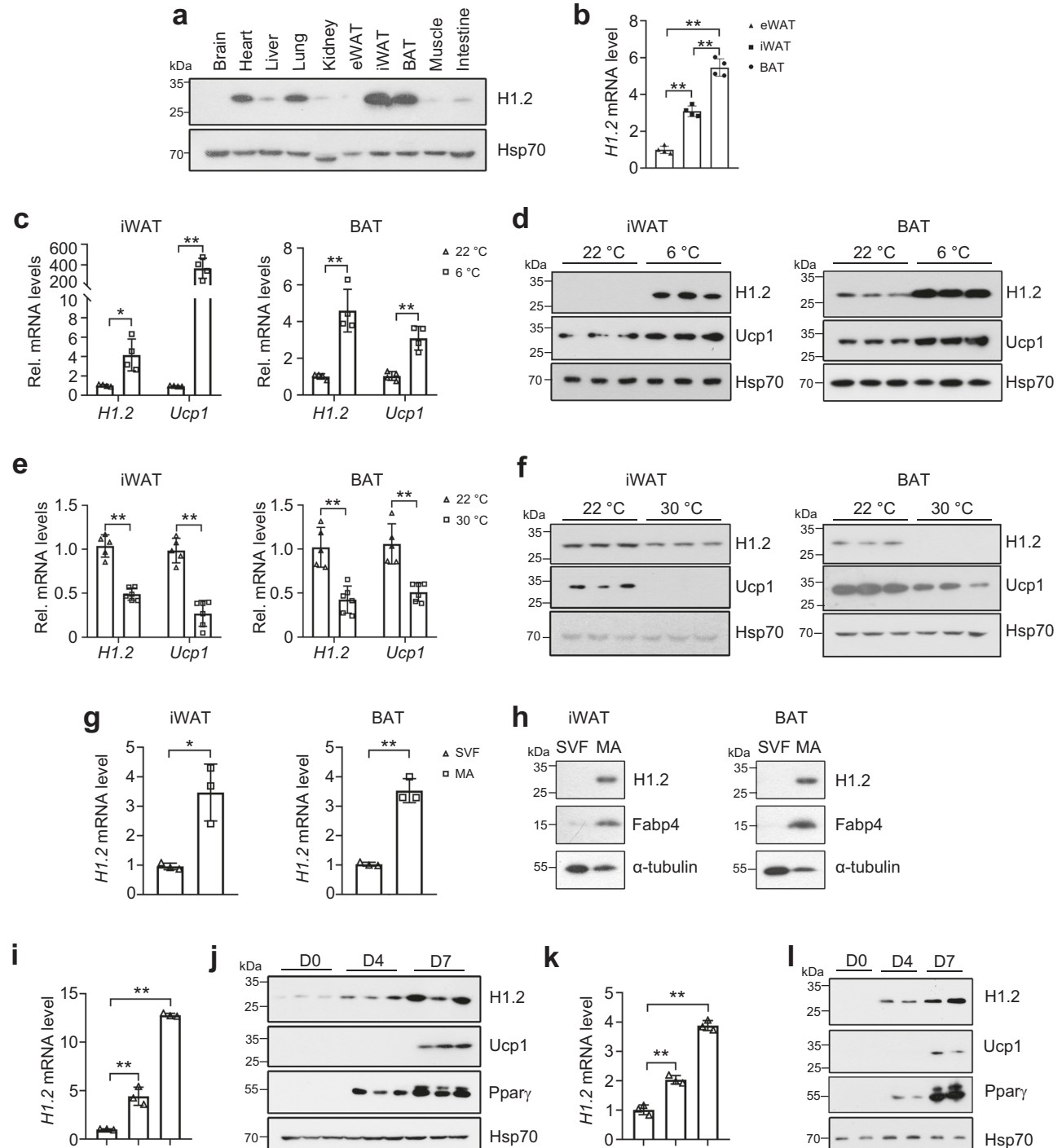

**Fig. 1 | H1.2 is enriched in beige/brown adipocytes and senses temperature change.** 10-week-old male mice were used in **a**–**d** and **g**–**l**; while 14-week-old male mice were used in **e** and **f**. **a** Representative mouse tissue expression profile of H1.2 (pooled samples from two individual mice for each lane; repeated three times independently with similar results obtained). **b** Relative *H1.2* mRNA level in different mouse fat pads (n = 4 per group; one-way ANOVA with Tukey's multiple comparisons test). $P_{H1.2 \, (eWAT \, vs \, iWAT)} < 0.0001$, $P_{H1.2 \, (eWAT \, vs \, BAT)} < 0.0001$, $P_{H1.2 \, (iWAT \, vs \, BAT)} < 0.0001$. **c**, **d** *H1.2* and *Ucp1* mRNA (**c**) and protein (**d**) levels in mouse iWAT and BAT at 22 °C or for 3 days at 6 °C (n = 4 per group in **c**; n = 3 per group in **d**; unpaired two-tailed Student's t test. **e** *H1.2* and *Ucp1* mRNA levels in mouse iWAT and BAT at 22 °C or for 4-week at 30 °C (22 °C group, n = 5; 30 °C group, n = 6; unpaired two-tailed Student's t test). **f** H1.2 and Ucp1 protein levels in mouse iWAT

and BAT at 22 °C or for 4-week at 30 °C (n = 3 per group). **g** Relative *H1.2* mRNA level in stromal vascular fraction (SVF) and mature adipocytes (MA) (n = 3; unpaired two-tailed Student's t test; repeated three times independently with similar results). **h** Representative H1.2 protein level in SVF and MA (pooled samples from three individual mice for each lane; repeated three times independently with similar results obtained). **i**, **j** mRNA (**i**) and protein (**j**) levels of H1.2 during primary beige adipocytes differentiation at indicated time (n = 3 per group; unpaired two-tailed Student's t-test). **k**, **l** Representative H1.2 mRNA (**k**) and protein (**l**) level during primary brown adipocytes differentiation at indicated time (n = 3 per time in **k**; n = 2 per time in **l**; unpaired two-tailed Student's t-test; repeated three times independently with similar results obtained). Data are mean ± S.D. *P < 0.05, **P < 0.01. Source data and exact P values are provided in a Source data file.

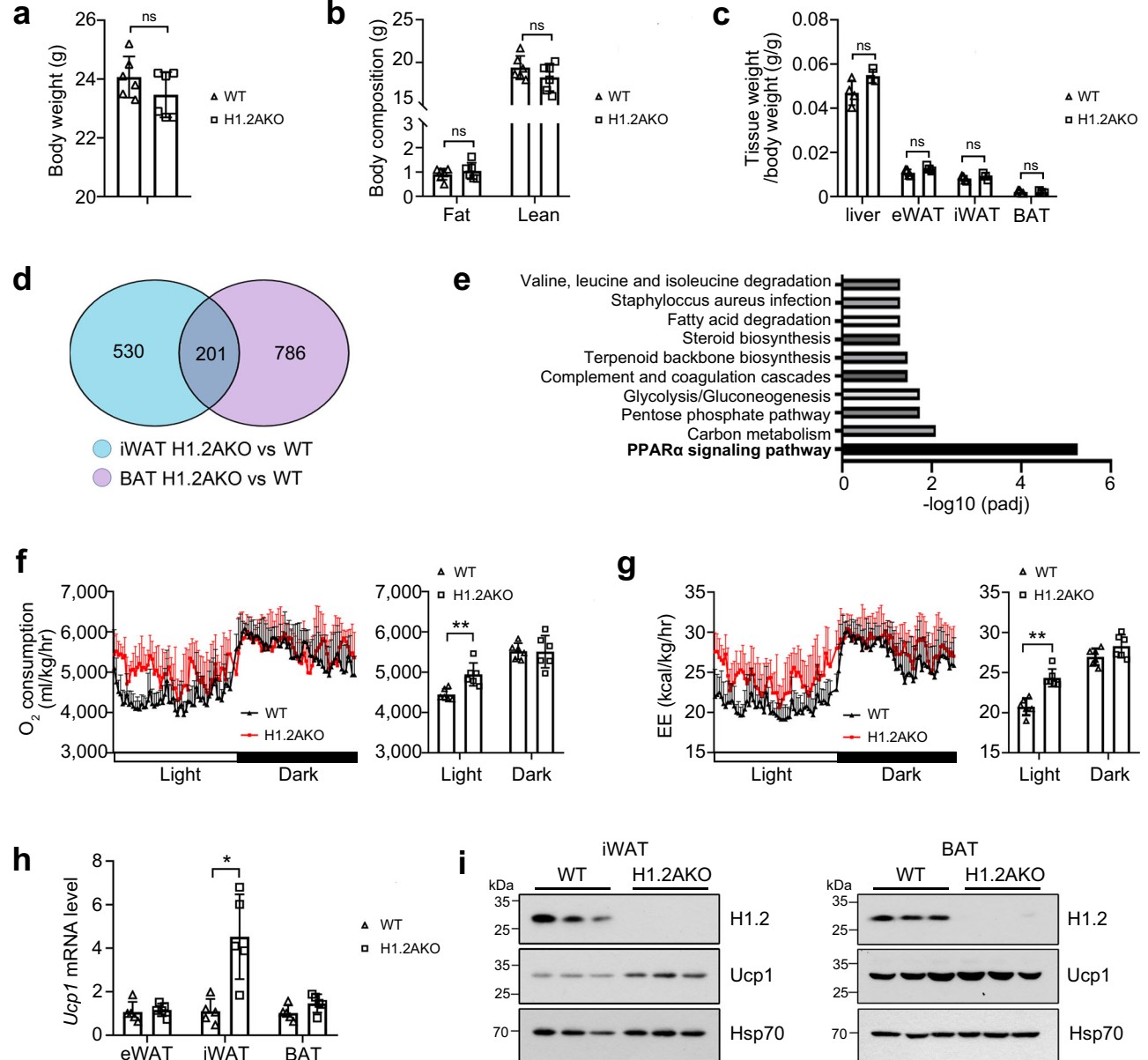

**Fig. 2 | Normal chow-fed young H1.2AKO mice show enhanced energy expenditure.** 10-week-old male mice were used in this figure. **a** Body weight of wildtype (WT) and H1.2AKO mice (*n* = 6 per group; unpaired two-tailed Student's *t* test). **b** Fat and lean mass of WT and H1.2AKO mice (*n* = 6 per group; unpaired two-tailed Student's *t* test). **c** Different tissue weights of WT and H1.2AKO mice (WT mice, *n* = 4; H1.2AKO mice, *n* = 3; unpaired two-tailed Student's *t* test). **d, e** Venn diagram (**d**) and KEGG pathway analysis (**e**) of differentially expressed genes in iWAT and BAT of WT and H1.2AKO mice (two samples per group, each sample was a pool of

two individual mice). **f, g** Oxygen consumption (**f**) and energy expenditure (EE) (**g**) with quantitative results of WT and H1.2AKO mice (*n* = 6 per group; two-tailed ANCOVA with body weight as a covariate). **h** *Ucp1* mRNA level in eWAT/iWAT/BAT of WT and H1.2AKO mice (*n* = 5 per group; unpaired two-tailed Student's *t* test). **i** Ucp1 and H1.2 protein levels in iWAT and BAT of WT and H1.2AKO mice (*n* = 3 per group). Data are mean ± S.D. *$P < 0.05$, **$P < 0.01$; ns not significant. Source data and exact *P* values are provided in a Source data file.

body weight (Supplementary Fig. 7a), but repressed $O_2$ consumption and energy expenditure during light (resting) phase in normal conditions (Supplementary Fig. 7b, c), which was consistent with those of young H1.2AKO mice (Fig. 2f, g). Consistently, after housing at 6 °C for 3 days, H1.2OE mice showed decreased cold tolerance as demonstrated by lower core body temperature compared with those of the control mice (Fig. 3h). Histological examination also revealed inhibited browning with reduced Ucp1 staining in iWAT of H1.2OE mice after cold exposure (Fig. 3i, j). Furthermore, decreased transcriptional levels of thermogenic genes, with unaffected transcriptional levels of other somatic H1 variants, were found in iWAT of H1.2OE mice after cold challenge (Fig. 3k and Supplementary Fig. 7d).

### Adipocyte H1.2 deficiency represses Il10rα level in iWAT

Since H1.2 in iWAT regulated cold-induced browning, differentially expressed genes in iWAT of H1.2AKO mice were further analyzed to identify factors that may responsible for this phenotype. Surprisingly, GO (gene ontology) analysis implicated significantly altered biological processes including humoral immune response (Fig. 4a), indicating that adipocyte H1.2 may regulate thermogenic remodeling through affecting immune cells in iWAT. Thus, several biomarkers of B cell, T cell, and chemokines associated with these cells were examined by qPCR. No change was found in B cell surface marker *Cd22* and *Cd45*, as well as in T cell surface marker *Cd3*, indicating that maturation and the number of immune cells related to humoral immune response were

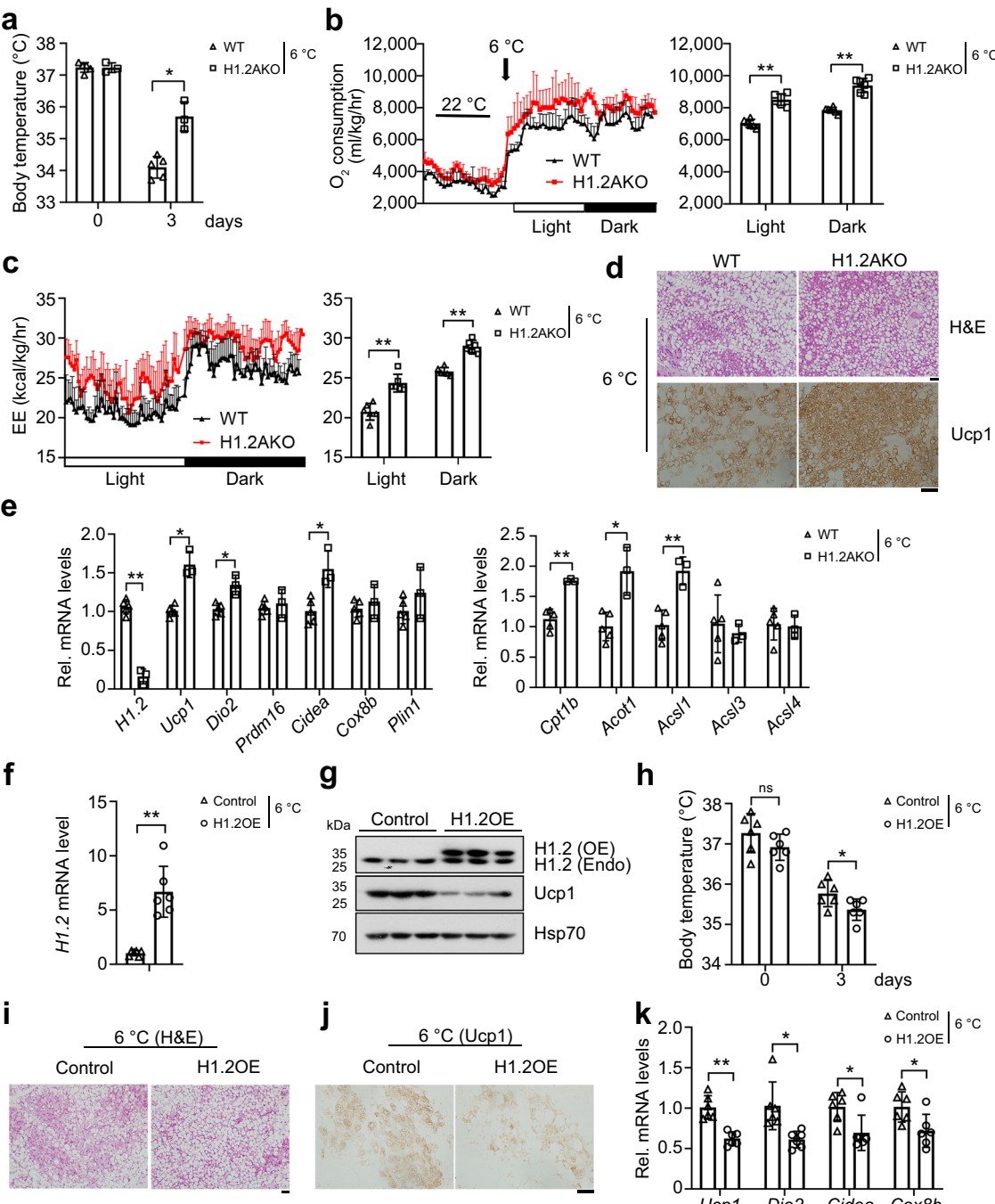

**Fig. 3 | H1.2 regulates iWAT browning under cold exposure.** In all, 10–11 weeks-old male mice were used in this figure. **a** Rectal temperature of WT and H1.2AKO mice before and after cold exposure (6 °C for 3 days; WT mice, $n = 5$; H1.2AKO mice, $n = 3$; unpaired two-tailed Student's $t$ test). **b**, **c** Oxygen consumption (**b**) and energy expenditure (EE) (**c**) with quantitative results of WT and H1.2AKO mice upon cold exposure ($n = 6$ per group; two-tailed ANCOVA with body weight as a covariate). **d** Representative images of H&E and Ucp1 staining of iWAT upon cold exposure ($n = 6$ per group). Scale bar = 50 μm. **e** qPCR analysis of thermogenic genes and lipid β-oxidation genes in iWAT of WT and H1.2AKO mice upon cold exposure (WT mice, $n = 5$; H1.2AKO mice, $n = 3$; unpaired two-tailed Student's $t$ test). **f** *H1.2* mRNA level in iWAT injected with AAV-Vehicle (control) or AAV-H1.2

(H1.2OE) after cold exposure ($n = 6$ per group; unpaired two-tailed Student's $t$ test). **g** H1.2 and Ucp1 protein levels in iWAT of control and H1.2OE mice upon cold exposure ($n = 3$ per group). **h** Rectal temperature of control and H1.2OE mice before and after cold exposure ($n = 6$ per group; unpaired two-tailed Student's $t$ test). **i**, **j** Representative images of H&E (**i**) and Ucp1 staining (**j**) in iWAT of control and H1.2OE mice after cold exposure ($n = 6$ per group). Scale bar = 50 μm. **k** qPCR analysis of thermogenic genes in iWAT of control and H1.2OE mice after cold exposure ($n = 6$ per group; unpaired two-tailed Student's $t$ test). Data are mean ± S.D. *$P < 0.05$, **$P < 0.01$. Source data and exact $P$ values are provided in a Source data file.

unaffected (Fig. 4b). However, in iWAT of H1.2AKO mice, we found significantly downregulated Il10 (Fig. 4b), a cytokine produced by multiple immune cells and reported to regulate beige adipocyte activation in a paracrine manner[8,9].

Il10 functions through a heterotetramer receptor complex, which consists of two Il10rα that bind Il10 and tissue-specifically initiate signaling, as well as two ubiquitously expressed Il10rβ which do not bind Il10, but sense the Il10-Il10rα binding[30]. Since both receptors exist in

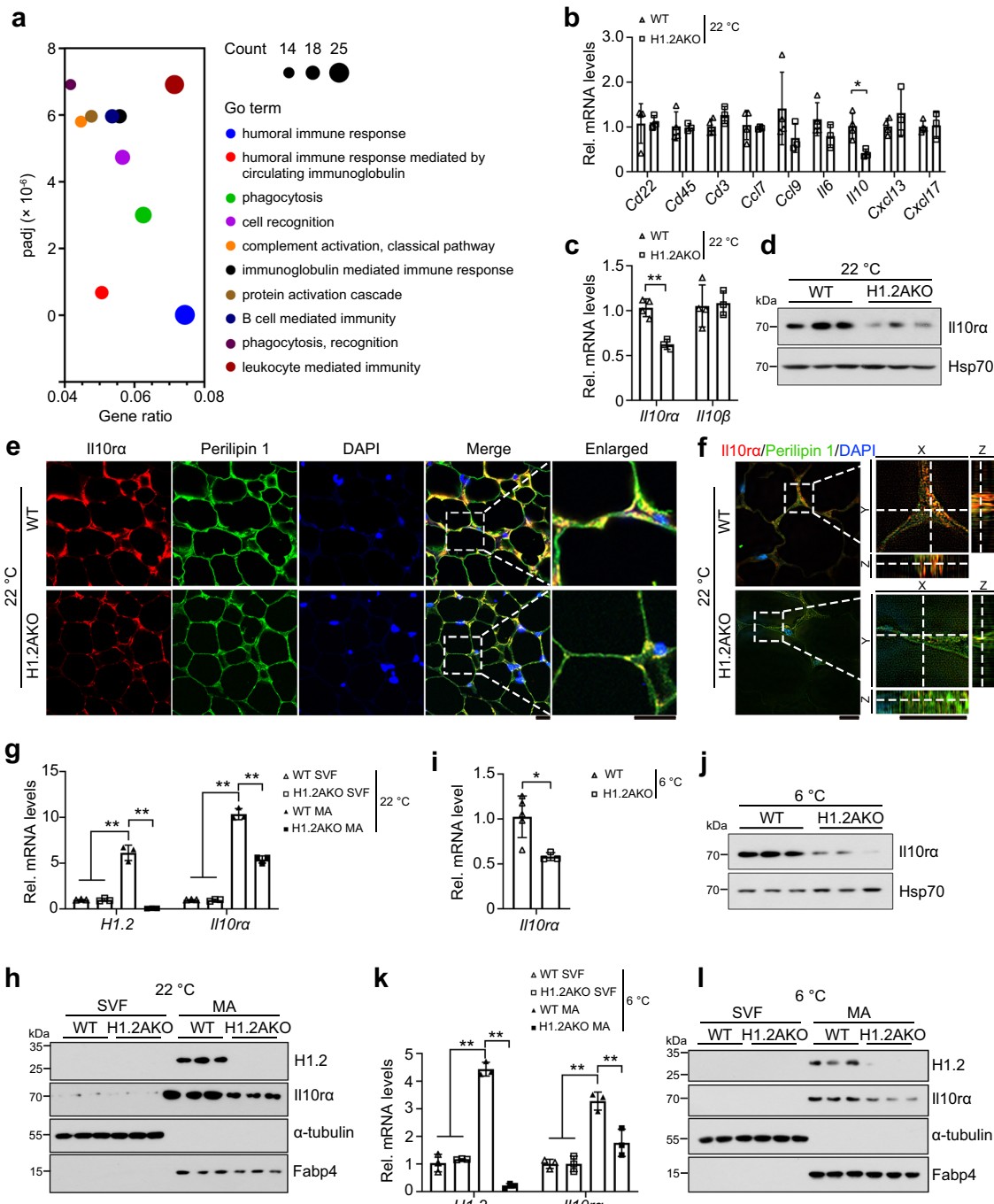

**Fig. 4 | Il10rα is downregulated in mature adipocytes of H1.2AKO mice.** 10-week-old male mice were used in this figure. **a** Enrichment of differentially expressed genes in iWAT of H1.2AKO mice (two samples for each group, each sample was a pool of two individual mice). **b**, **c** qPCR analysis of genes related to humoral immune pathways (**b**), and *Il10rα* or *Il10rβ* (**c**) in iWAT of WT and H1.2AKO mice (WT mice, $n = 4$; H1.2AKO mice, $n = 3$). **d**, **e** Il10rα protein level (**d**) and representative immunofluorescence staining of Il10rα (red) and Perilipin 1 (green) (**e**) in iWAT of WT and H1.2AKO mice ($n = 3$ per group). Scale bar = 20 μm. **f** Representative images of colocalized Il10rα (red) and Perilipin 1 (green) in iWAT of WT and H1.2AKO mice; magnified orthogonal sectioning view of regions in the insert boxes ($n = 3$ per group). Scale bar = 10 μm. **g**, **h** H1.2 and Il10rα mRNA (**g**) and protein levels (**h**) in stromal vascular fraction (SVF) and mature adipocytes (MA) fractionated from iWAT of WT and H1.2AKO mice ($n = 3$ per group). In **g**: $P_{H1.2/Il10rα}$ (WT SVF vs WT MA) < 0.0001, $P_{H1.2/Il10rα}$ (H1.2AKO SVF vs WT MA) < 0.0001, $P_{H1.2/Il10rα}$ (WT MA vs H1.2AKO MA) < 0.0001. **i** *Il10rα* mRNA level in iWAT of WT and H1.2AKO mice after cold exposure (WT mice, $n = 5$; H1.2AKO mice, $n = 3$). **j** Il10rα protein level in iWAT of WT and H1.2AKO mice after cold exposure ($n = 3$ per group). **k**, **l** H1.2 and Il10rα mRNA (**k**) and protein (**l**) levels in SVF and MA fractionated from iWAT of WT and H1.2AKO mice after cold exposure ($n = 3$ per group). In **k**: $P_{H1.2}$ (WT SVF vs WT MA) < 0.0001, $P_{H1.2}$ (H1.2AKO SVF vs WT MA) < 0.0001, $P_{H1.2}$ (WT MA vs H1.2AKO MA) < 0.0001; $P_{Il10rα}$ (WT SVF vs WT MA) = 0.0002, $P_{Il10rα}$ (H1.2AKO SVF vs WT MA) = 0.0002, $P_{Il10rα}$ (WT MA vs H1.2AKO MA) = 0.0026. Data are mean ± S.D. Unpaired two-tailed Student's $t$ test in **b**, **c** and **i**; one-way ANOVA with Tukey's multiple comparisons test in **g** and **k**. *$P < 0.05$, **$P < 0.01$. Source data and exact $P$ values are provided in a Source data file.

adipocytes[10], the levels of Il10rα and Il10rβ in iWAT were first examined. Significantly reduced mRNA level of Il10rα, but not Il10rβ, was found in iWAT of H1.2AKO mice under normal conditions (Fig. 4c). Consistently, the protein level of Il10rα was also reduced in iWAT of H1.2AKO mice (Fig. 4d). Immunofluorescence staining further confirmed that H1.2 deficiency greatly reduced Il10rα expression in adipocytes of iWAT (Fig. 4e, f). The level of Il10rα was also examined in mature adipocytes and SVF isolated from iWAT of WT and H1.2AKO mice under normal conditions. Higher levels of H1.2 and Il10rα were found in mature beige adipocytes of WT mice; while decreased Il10rα levels were found in mature beige adipocytes of H1.2AKO mice (Fig. 4g, h). After cold stimulation, downregulated Il10rα levels were also found in iWAT, especially in mature beige adipocytes, of H1.2AKO mice (Fig. 4i–l).

In H1.2AKO mice, GO pathway analysis of RNA-seq in BAT showed a different enrichment of altered biological processes compared with those in iWAT (Supplementary Fig. 8a). Furthermore, under normal or cold exposure conditions, neither the transcriptional levels of genes related to humoral immune response, nor the transcriptional levels of Il10 and Il10rα/Il10rβ, was changed in BAT of H1.2AKO mice (Supplementary Fig. 8b–f). These data indicated that adipocyte H1.2 deficiency represses Il10rα expression specifically in iWAT.

## H1.2 regulates thermogenesis in primary beige adipocytes via Il10rα

Next, we explored whether H1.2 affected thermogenesis in a cell autonomous manner. SVF cells isolated from iWAT or BAT of *H1.2^flox/flox* mice were differentiated into mature beige or brown adipocytes, respectively; and *H1.2* knockout was achieved by infecting adenovirus-packed Cre recombinases (Fig. 5a and Supplementary Fig. 9a). Notably, depletion of H1.2 in mature beige adipocytes or brown adipocytes did not affect adipocyte differentiation and lipid accumulation, as well as transcriptional levels of key adipogenic genes, including *Adipoq, Fabp4* (fatty acid binding protein 4) and *Pparγ* (peroxisome proliferator activated receptor gamma; Fig. 5b, c and Supplementary Fig. 9b, c). However, transcriptional levels of thermogenic genes, such as *Ucp1, Dio2, Cidea*, and *Cox8b (cytochrome c oxidase, subunit VIIIb)*, as well as lipid β-oxidation related genes, such as *Ppara* (peroxisome proliferator-activated receptor alpha), *Acox1* (acyl-Coenzyme A oxidase 1), *Acsl1* and *Cpt1b*, were increased in H1.2 deficient primary beige adipocytes (Fig. 5d, e). Meanwhile, the protein level of Ucp1 was upregulated in H1.2 deficient primary beige adipocytes (Fig. 5f). In contrast, transcriptional levels of *Ucp1, Dio2, Cidea*, and *Cox8b*, as well as the protein level of Ucp1, were unchanged in H1.2 deficient primary brown adipocytes (Supplementary Fig. 9d, e). These data indicated that H1.2 was crucial for the thermogenic capacity of beige adipocytes, but dispensable for maintenance of beige and brown adipocyte features.

We next examined whether H1.2 inhibited beige adipocyte thermogenesis via regulating Il10rα in vitro. Il10rα levels were decreased in differentiated H1.2 deficient beige adipocytes, with or without isoproterenol treatment (a β-adrenergic receptor agonist to stimulate thermogenesis[31]) (Fig. 5g, h). To examine whether Il10-Il10rα signaling-mediated thermogenesis was affected in H1.2 deficiency beige adipocytes, recombinant mouse Il10 was used. Decreased mRNA levels of *Ucp1, Dio2, Cidea*, and *Cox8b*, as well as Ucp1 protein level were found in Il10-treated differentiated beige adipocytes; however, upon knockout of H1.2, this thermogenic inhibitory effect of Il10 was attenuated, associated with decreased Il10rα levels (Fig. 5i, j). Furthermore, overexpression of Il10rα downregulated transcriptional levels of thermogenesis-related genes (*Ucp1* and *Dio2*), as well as protein level of Ucp1 in H1.2 knockout beige adipocytes treated with Il10 (Fig. 5k–m). These results suggested that H1.2 plays an important role in Il10-Il10rα signaling-mediated thermogenesis in beige cells.

To investigate whether H1.2 directly binds to the promoter of *Il10rα*, chromatin immunoprecipitation (ChIP) assay was performed. H1.2 bound to the promoter of *Il10rα*, but not the promoter of *Gapdh*, in iWAT of WT mice, but this binding was abolished in H1.2AKO mice (Supplementary Fig. 10a). Moreover, consistent with cold-induced H1.2 upregulation (Fig. 1c, d), cold challenge promoted the binding of H1.2 on *Il10rα* promoter in iWAT of WT mice, while the binding of H1.2 on *Nrp2* (recombinant neuropilin 2) promoter was unchanged (Supplementary Fig. 10b). Luciferase reporter assay was further performed to study the transcriptional regulation of full-length and different domain-deleted H1.2 on *Il10rα* expression in HEK293T cells. Knockdown of H1.2 downregulated *Il10rα* transcription, while overexpression of H1.2 increased *Il10rα* transcription, predominantly depending on its globular or C-terminal domain (Supplementary Fig. 10c–e). Moreover, among several clinical *H1.2* mutations identified by cancer genome sequencing[23,32], mutations P118S and A171P affected the transcriptional level of *Il10rα* (Supplementary Fig. 10f).

To investigate the chromatin distribution and additional potential roles of H1.2 in iWAT function, chromatin immunoprecipitation sequencing (ChIP-seq) was performed in iWAT of C57BL/6 mice under normal conditions and upon cold exposure. Overall, 10,418 and 12,547 genes were detected under normal conditions or upon cold exposure, respectively. Of these genes, more than 50% of H1.2 binding sites were found in intergenic regions, while 13-15% were found in promoters and transcription start sites (Supplementary Fig. 11a), which were similar to those found in histone H3.3 or H3K9me3 ChIP-seq assay[33,34]. Furthermore, cold stimulation caused higher intensity of H1.2 peaks on genomes, especially on promoter regions (Supplementary Fig. 11a, b). GO analysis revealed a significant H1.2 enrichment on genes in biological processes including lipid metabolism, oxidative phosphorylation, and immune response after cold stimulation (Supplementary Fig. 11c). Meanwhile, stronger H1.2 binding on *Il10rα* gene upon cold exposure was also detected (Supplementary Fig. 11d).

## H1.2 affects cold-induced iWAT browning through regulating Il10rα

To investigate whether downregulated Il10rα is responsible for H1.2 deficiency enhanced browning under cold stimulation, Il10rα was overexpressed in WT and H1.2AKO mice by in situ injection of AAV-Il10rα in the left iWAT pad, with the control AAV injected into the right iWAT pad (Fig. 6a). Mice were challenged with cold at three weeks after injection. Significantly overexpressed Il10rα was found in AAV-Il10rα injected iWAT, without affecting iWAT weights, after cold stimulation (Fig. 6b, c and Supplementary Fig. 12a). Overexpression of Il10rα showed no obvious effect on the transcriptional levels of pro-inflammatory or anti-inflammatory cytokines examined in iWAT of WT or H1.2AKO mice (Supplementary Fig. 12b). However, overexpression of Il10rα reduced cold-induced iWAT browning in WT mice, as demonstrated by histological analysis, Ucp1 staining, and transcriptional levels of thermogenic genes; moreover, Il10rα overexpression completely abrogated the enhanced iWAT browning in cold-challenged H1.2AKO mice (Fig. 6d–g). Collectively, these data demonstrated that H1.2 suppressed cold-induced beige adipocyte thermogenesis via Il10rα.

## H1.2AKO mice have better metabolic status after long-term NC-feeding

Involvement of adipocyte H1.2 in thermogenesis enables us to hypothesize that H1.2AKO mice may attenuate fat accumulation under physiological or pathological conditions. Although comparable body weights were found between NC-fed young WT and H1.2AKO mice (Fig. 2a), they started to show difference in body weight at 18-week-old, which became more obvious at 28-week-old, with H1.2AKO mice gained less fat mass and lean mass (Fig. 7a, b). Moreover, anatomical results showed reduced eWAT and BAT

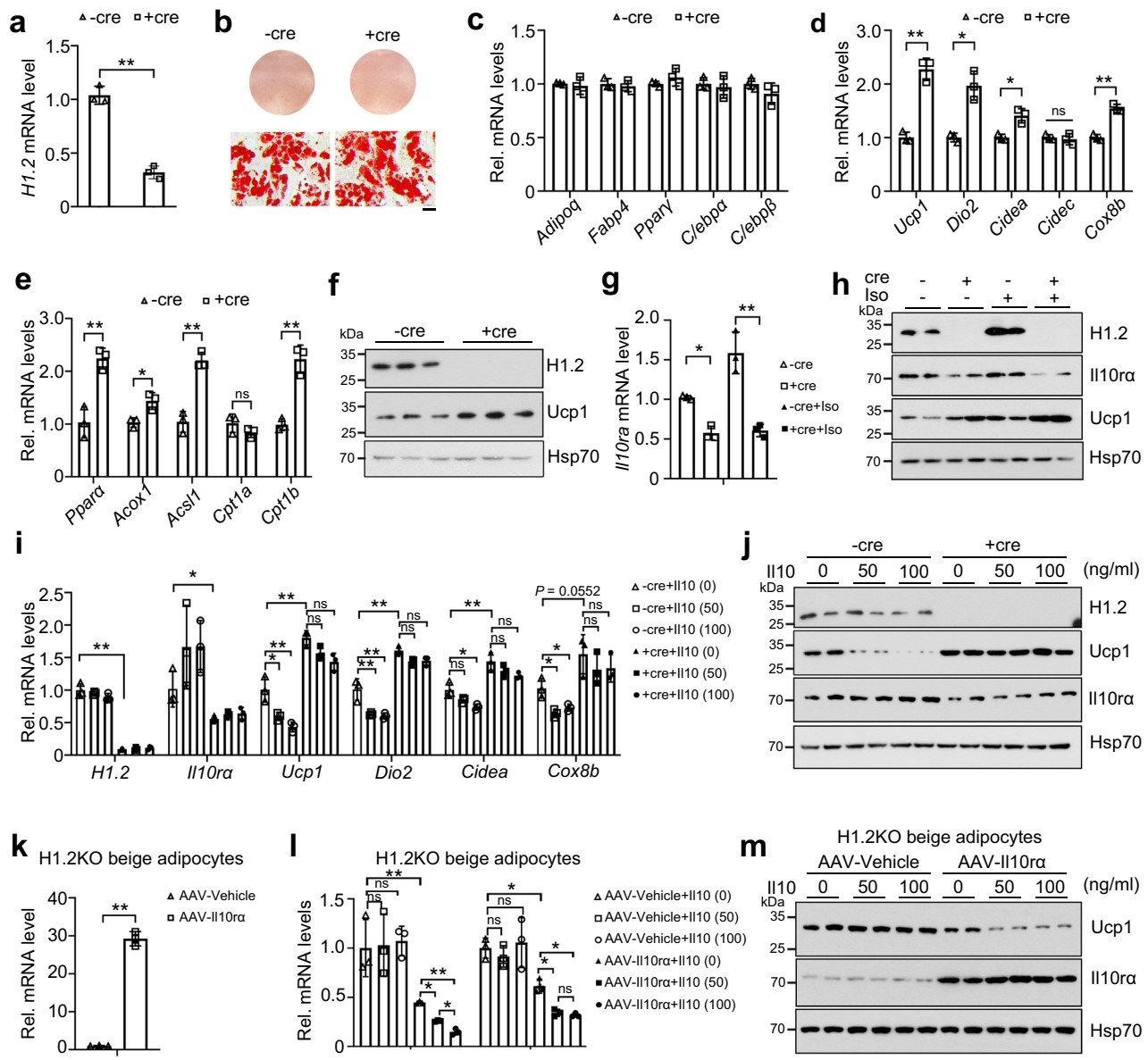

**Fig. 5 | H1.2 regulates Il10rα and thermogenesis in beige adipocytes in vitro.**
**a**, **b** *H1.2* mRNA level (**a**) and representative Oil red O staining (**b**) in differentiated beige adipocytes isolated from *H1.2flox/flox* mice treated with adenovirus-packed Cre recombinase ( + Cre) or vehicles (-Cre) (*n* = 3, repeated three times independently with similar results obtained). Scale bar = 50 μm. **c–f** mRNA levels of genes related to adipogenesis (**c**), thermogenesis (**d**) and lipid β-oxidation (**e**), and protein levels of H1.2 and Ucp1 (**f**) of differentiated beige adipocytes isolated from *H1.2flox/flox* mice treated with or without Cre (*n* = 3). **g**, **h** *Il10rα* mRNA level (**g**) and representative Il10rα and Ucp1 protein levels (**h**) in differentiated beige adipocytes isolated from *H1.2flox/flox* mice, with or without Cre/10 μM isoprenaline (Iso) treatments (*n* = 3 in **g**; *n* = 2 in **h**; repeated three times independently with similar results obtained). **i** qPCR of thermogenic genes of differentiated beige adipocytes isolated from *H1.2flox/flox* mice, with or without Cre/Il10 treatments (*n* = 3). $P_{H1.2}$ (-cre+Il10-0 vs -cre+Il10-50) = 0.8879, $P_{H1.2}$ (-cre+Il10-0 vs -cre+Il10-100) = 0.1621, $P_{H1.2}$ (-cre+Il10-0 vs +cre+Il10-0) < 0.0001; $P_{Il10ra}$ (-cre+Il10-0 vs -cre+Il10-50) = 0.2056, $P_{Il10ra}$ (-cre+Il10-0 vs -cre+Il10-100) = 0.1964, $P_{Il10ra}$ (-cre+Il10-0 vs +cre+Il10-0) = 0.0433; $P_{Ucp1}$(-cre+Il10-0 vs -cre+Il10-50) = 0.0105, $P_{Ucp1}$ (-cre+Il10-0 vs -cre+Il10-100) = 0.0007, $P_{Ucp1}$ (-cre+Il10-0 vs +cre+Il10-0) < 0.0001, $P_{Ucp1}$ (+cre+Il10-0 vs +cre+Il10-50) = 0.4609, $P_{Ucp1}$ (+cre+Il10-0 vs +cre+Il10-100) = 0.0639; $P_{Dio2}$(-cre+Il10-0 vs -cre+Il10-50) = 0.0005, $P_{Dio2}$ (-cre+Il10-0 vs -cre+Il10-100) = 0.0002, $P_{Dio2}$ +cre+Il10-0 vs +cre+Il10-0) < 0.0001, $P_{Dio2}$ (+cre+Il10-0 vs +cre+Il10-50) = 0.1604, $P_{Dio2}$ (+cre+Il10-0 vs +cre+Il10-100) = 0.2028; $P_{Cidea}$ (-cre+Il10-0 vs -cre+Il10-50) = 0.3045, $P_{Cidea}$ (-cre+Il10-0 vs -cre+Il10-100) = 0.0276, $P_{Cidea}$ (-cre+Il10-0 vs +cre+Il10-0) = 0.0005,

$P_{Cidea}$ (+cre+Il10-0 vs +cre+Il10-50) = 0.3547, $P_{Cidea}$ (+cre+Il10-0 vs +cre+Il10-100) = 0.0737; $P_{Cox8b}$ (-cre+Il10-0 vs -cre+Il10-50) = 0.0296, $P_{Cox8b}$ (-cre+Il10-0 vs -cre+Il10-100) = 0.0236, $P_{Cox8b}$ (-cre+Il10-0 vs +cre+Il10-0) = 0.0552, $P_{Cox8b}$ (+cre+Il10-0 vs +cre+Il10-50) = 0.6737, $P_{Cox8b}$ (+cre+Il10-0 vs +cre+Il10-100) = 0.7483. **j** Ucp1 and Il10rα levels of differentiated beige adipocytes isolated from *H1.2flox/flox* mice, with or without Cre/Il10 treatments. **k** *Il10rα* level in differentiated beige adipocytes isolated from H1.2AKO mice, with or without AAV-Il10rα treatments (*n* = 3; repeated three times independently with similar results obtained). **l** *Ucp1* and *Dio2* levels in differentiated beige adipocytes isolated from H1.2AKO mice, with or without AAV-Il10rα/Il10 treatments (*n* = 3). $P_{Ucp1}$(AAV-Vehicle+Il10-0 vs AAV-Vehicle+Il10-50) = 0.9997, $P_{Ucp1}$ (AAV-Vehicle+Il10-0 vs AAV-Vehicle+Il10-100) = 0.9839, $P_{Ucp1}$ (AAV-Vehicle+Il10-0 vs AAV-Il10rα+Il10-0) = 0.0043, $P_{Ucp1}$ (AAV-Il10rα+Il10-0 vs AAV-Il10rα+Il10-50) = 0.0467, $P_{Ucp1}$ (AAV-Il10rα+Il10-0 vs AAV-Il10rα+Il10-100) = 0.0019, $P_{Ucp1}$ (AAV-Il10rα+Il10-50 vs AAV-Il10rα+Il10-100) = 0.0141; $P_{Dio2}$ (AAV-Vehicle+Il10-0 vs AAV-Vehicle+Il10-50) = 0.8793, $P_{Dio2}$ (AAV-Vehicle+Il10-0 vs AAV-Vehicle+Il10-100) = 0.9737, $P_{Dio2}$ (AAV-Vehicle+Il10-0 vs AAV-Il10rα+Il10-0) = 0.0104, $P_{Dio2}$ (AAV-Il10rα+Il10-0 vs AAV-Il10rα+Il10-50) = 0.0451, $P_{Dio2}$ (AAV-Il10rα+Il10-0 vs AAV-Il10rα +Il10-100) = 0.0228, $P_{Dio2}$ (AAV-Il10rα+Il10-50 vs AAV-Il10rα+Il10-100) = 0.9999. **m** Ucp1 and Il10rα levels of differentiated beige adipocytes isolated from H1.2AKO mice, with or without AAV-Il10rα/Il10 treatments (*n* = 2; repeated three times independently with similar results). Data are mean ± S.D. Unpaired two-tailed Student's *t* test in **a**, **c–f**, **k**; one-way ANOVA with Tukey's multiple comparisons test in **g**, **i** and **l**. *P < 0.05, **P < 0.01; ns not significant. Source data and exact P values are provided in a Source data file.

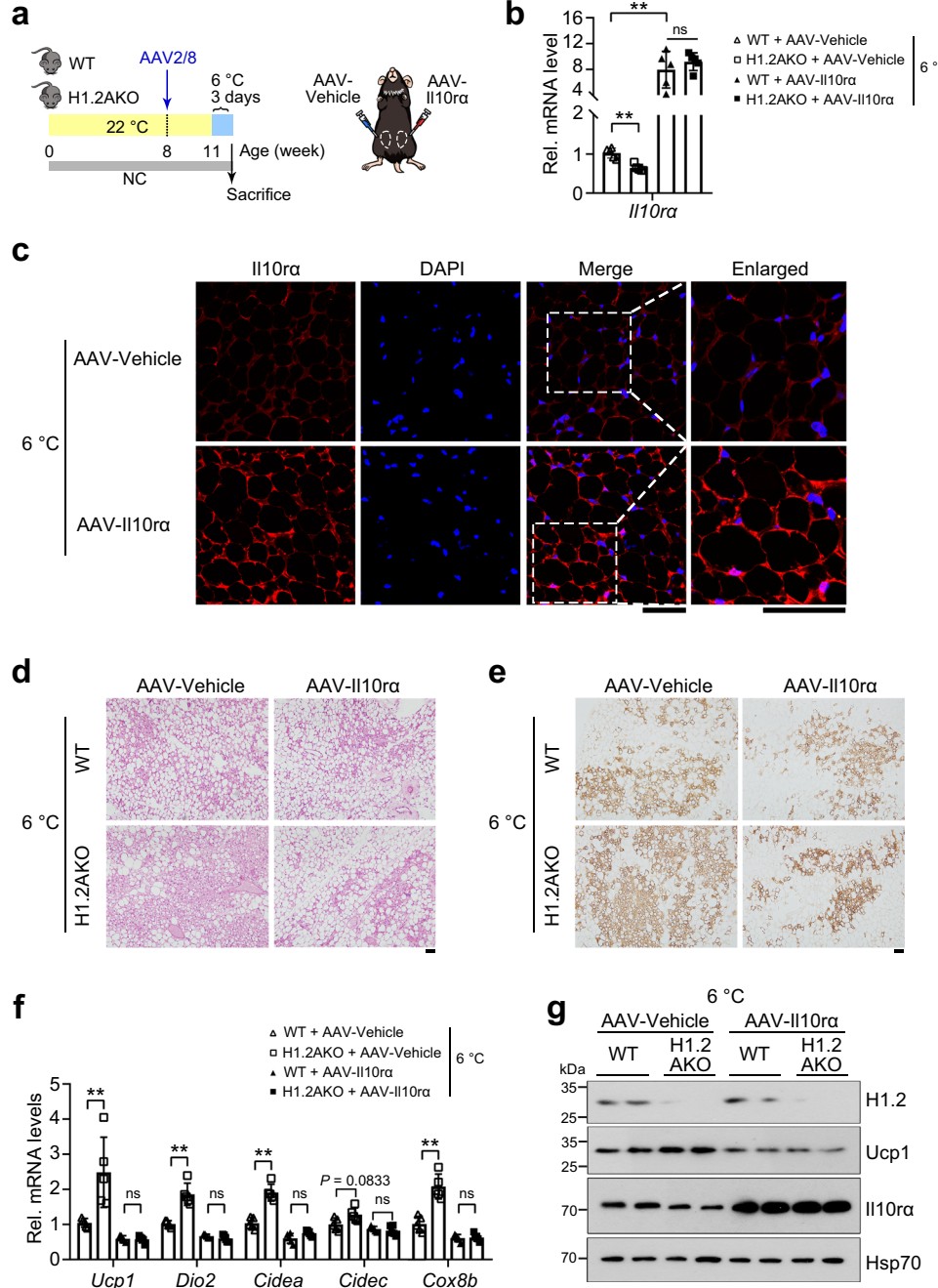

**Fig. 6 | H1.2 regulates cold-induced iWAT browning through Il10rα.**
**a** Experimental design. WT and H1.2AKO mice were injected with AAV-Il10rα (left iWAT pad) and AAV-Vehicle (right iWAT pad) at 8-week-old, 3 weeks later, then exposed under 6 °C for 3 days. **b** *Il10rα* mRNA level in iWAT after cold exposure (6 °C for 3 days) in the indicated groups ($n = 5$ per group; one-way ANOVA with Tukey's multiple comparisons test). $P_{Il10rα}$ (WT+AAV-Vehicle vs H1.2AKO+AAV-Vehicle) < 0.0001, $P_{Il10rα}$ (WT+AAV-Vehicle vs WT+AAV-Il10rα) < 0.0001, $P_{Il10rα}$ (WT+AAV-Il10rα vs H1.2AKO+AAV-Il10rα) = 0.6088. **c** Representative Il10rα staining of iWAT injected with AAV-Il10rα or AAV-Vehicle after cold exposure ($n = 5$ per group). Scale bar = 50 μm. **d, e** Representative H&E (**d**) and Ucp1 staining (**e**) of iWAT in WT and H1.2AKO mice injected with AAV-Il10rα or AAV-Vehicle after cold exposure ($n = 5$ per group). Scale bar = 50 μm. **f** mRNA levels of thermogenic genes in iWAT of WT and H1.2AKO mice injected with

AAV-Il10rα or AAV-Vehicle after cold exposure ($n = 5$ per group; one-way ANOVA with Tukey's multiple comparisons test). $P_{Ucp1}$ (WT+AAV-Vehicle vs H1.2AKO+AAV-Vehicle) = 0.0019, $P_{Ucp1}$ (WT+AAV-Il10rα vs H1.2AKO+AAV-Il10rα) > 0.9999, $P_{Dio2}$ (WT+AAV-Vehicle vs H1.2AKO+AAV-Vehicle) < 0.0001, $P_{Dio2}$ (WT+AAV-Il10rα vs H1.2AKO+AAV-Il10rα) = 0.9264, $P_{Cidea}$ (WT+AAV-Vehicle vs H1.2AKO+AAV-Vehicle) < 0.0001, $P_{Cidea}$ (WT+AAV-Il10rα vs H1.2AKO+AAV-Il10rα) = 0.6903, $P_{Cidec}$ (WT+AAV-Vehicle vs H1.2AKO+AAV-Vehicle) = 0.0833, $P_{Cidec}$ (WT+AAV-Il10rα vs H1.2AKO+AAV-Il10rα) = 0.8226, $P_{Cox8b}$ (WT+AAV-Vehicle vs H1.2AKO+AAV-Vehicle) < 0.0001, $P_{Cidea}$ (WT+AAV-Il10rα vs H1.2AKO+AAV-Il10rα) = 0.9988. **g** Representative H1.2, Ucp1 and Il10rα protein levels in iWAT of WT and H1.2AKO mice injected with AAV-Il10rα or AAV-Vehicle after cold exposure ($n = 3$ per group). Data are mean ± S.D. **$P < 0.01$; ns not significant. Source data and exact $P$ values are provided in a Source data file.

weights, but no change in iWAT and liver weights, as well as smaller adipocyte size in eWAT and iWAT of 28-week-old H1.2AKO mice under normal conditions (Fig. 7c–e). Importantly, in 28-week-old H1.2AKO mice under normal conditions, elevated $O_2$ consumption and energy expenditure, as well as improved glucose tolerance and

insulin sensitivity, were observed (Fig. 7f–i). Downregulated lipogenic genes, and upregulated thermogenic and lipid β-oxidation genes were found in iWAT of 28-week-old H1.2AKO mice (Fig. 7j); while upregulated Ucp1 and downregulated Il10rα were consistently observed (Fig. 7k).

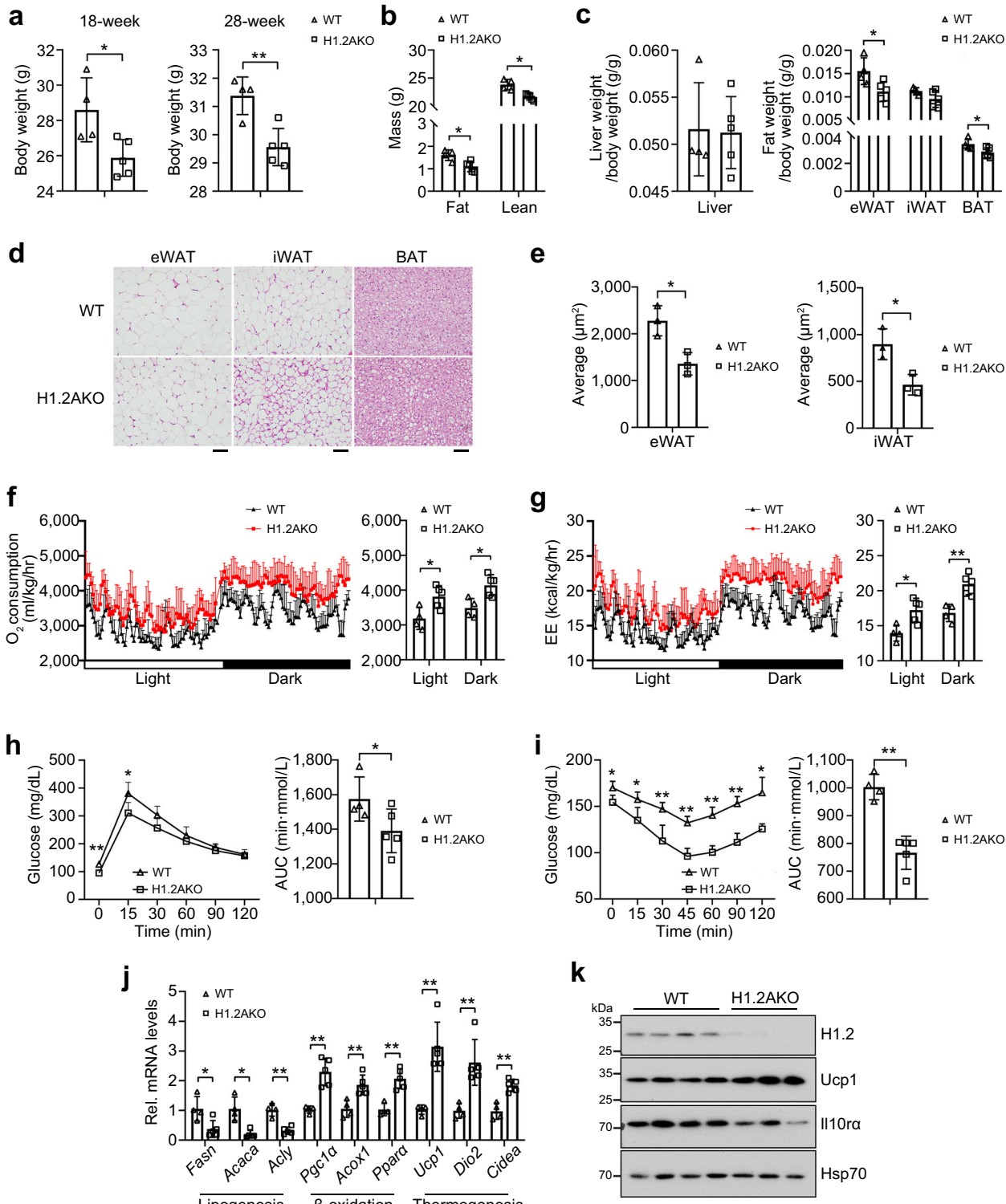

**Fig. 7 | H1.2AKO mice have a better metabolic status after long-term normal chow-feeding. a** Body weight of normal chow (NC)-fed WT and H1.2AKO mice at 18-week-old and 28-week-old (WT mice, $n = 4$; H1.2AKO mice, $n = 5$; unpaired two-tailed Student's $t$ test). **b, c** Fat mass or lean mass (**b**) and different tissue weights (**c**) of 28-week-old NC-fed WT and H1.2AKO mice (WT mice, $n = 4$; H1.2AKO mice, $n = 5$; unpaired two-tailed Student's $t$ test). **d** Representative H&E staining of eWAT/iWAT/BAT of 28-week-old NC-fed WT and H1.2AKO mice (WT mice, $n = 4$; H1.2AKO mice, $n = 5$). Scale bar = 50 μm. **e** Average adipocyte area of eWAT and iWAT of 28-week-old NC-fed WT and H1.2AKO mice ($n = 3$ per group; unpaired two-tailed Student's $t$ test). **f, g** Oxygen consumption (**f**) and energy expenditure (EE) (**g**) of 28-week-old NC-fed WT and H1.2AKO mice (WT mice, $n = 4$; H1.2AKO mice, $n = 5$; two-tailed ANCOVA with body weight as a covariate). **h, i** glucose tolerance test (**h**) and insulin tolerance test (**i**) of 28-week-old NC-fed WT and H1.2AKO mice (WT mice, $n = 4$; H1.2AKO mice, $n = 5$; unpaired two-tailed Student's $t$ test). **j** qPCR of genes associated with lipogenesis, lipid β-oxidation and thermogenesis in iWAT of 28-week-old NC-fed WT and H1.2AKO mice (WT mice, $n = 4$; H1.2AKO mice, $n = 5$; unpaired two-tailed Student's $t$ test). **k** H1.2, Ucp1, and Il10rα levels in iWAT of 28-week-old NC-fed WT and H1.2AKO mice (WT mice, $n = 4$; H1.2AKO mice, $n = 3$). Data are mean ± S.D. *$P < 0.05$, **$P < 0.01$. Source data and exact $P$ values are provided in a Source data file.

Recent studies suggested that housing in normal conditions (room temperature, 22 °C) presents a mild cold stimulation for mice compared to thermoneutral conditions (30 °C)[35]. To investigate whether adrenergic signaling-induced adaptive thermogenesis contributes to the metabolic phenotype of long-term NC-fed H1.2AKO mice, we housed mice at 30 °C for four months to minimize adaptive thermogenesis (Fig. 8a). The body weight, eWAT/iWAT/liver weights showed no difference, and lower BAT weight was found in H1.2AKO mice (Fig. 8b–e). Comparable adipocyte sizes of eWAT/iWAT/BAT, transcriptional levels of lipogenic, thermogenic, and lipid β-oxidation genes were found in iWAT of WT and H1.2AKO mice under thermoneutral conditions (Fig. 8f, g). These data indicated adaptive thermogenesis as a key player mediating the functions of adipocyte H1.2 on weight gain or adipocyte morphology.

## Deletion of H1.2 in adipocytes protects against HFD-induced obesity

Overnutrition is related to adipose dysfunction and thermogenesis impairment, which results in obesity and metabolic syndrome. We next examined whether HFD feeding affects H1.2 expression in thermogenic adipose tissues. After six months of HFD stress, the mRNA and protein levels of H1.2 were increased in both iWAT and BAT compared with those in NC-fed mice (Fig. 9a, b). To determine the clinical relevance between H1.2 and obesity, subcutaneous WATs were collected from human subjects with different body mass index (BMI).

Upregulated H1.2 protein level in subcutaneous WAT of obese subjects was found (Fig. 9c). Consistently, significantly increased mRNA level of *H1.2* was observed in subcutaneous WAT of obese subjects (Fig. 9d). Regression analysis revealed a positive correlation between BMI and the *H1.2* mRNA level in subcutaneous WAT (Fig. 9e). Therefore, we hypothesized that highly expressed H1.2 may result in thermogenic adipose tissue dysfunction during HFD-induced obesity.

After 21 weeks of HFD challenge, H1.2AKO mice showed resistance to obesity compared with WT mice, as indicated by significantly lower body weight, even with higher amount of food intake (Fig. 9f and Supplementary Fig. 13a, b). Reduced fat mass but not lean mass, as well as reduced iWAT and BAT weights, were observed in H1.2AKO mice under HFD stress (Fig. 9g and Supplementary Fig. 13c, d). Moreover, HFD-fed H1.2AKO mice showed elevated $O_2$ consumption and energy expenditure, and decreased adipocyte size of iWAT and eWAT (Fig. 9h, i and Supplementary Fig. 13e, f). Furthermore, HFD-fed H1.2AKO mice showed significantly decreased non-fasted blood glucose level (Supplementary Fig. 13g), as well as significantly improved glucose tolerance and insulin tolerance (Fig. 9j, k). Lipogenic genes were downregulated, whereas thermogenic and lipid β-oxidation genes were significantly upregulated in iWAT of HFD-fed H1.2AKO mice (Fig. 9l). Consistently, increased Ucp1 and reduced Il10rα level were found in iWAT of HFD-fed H1.2AKO mice (Fig. 9m). These results suggested that adipocyte H1.2 ablation elicits metabolic benefits against overnutrition caused obesity.

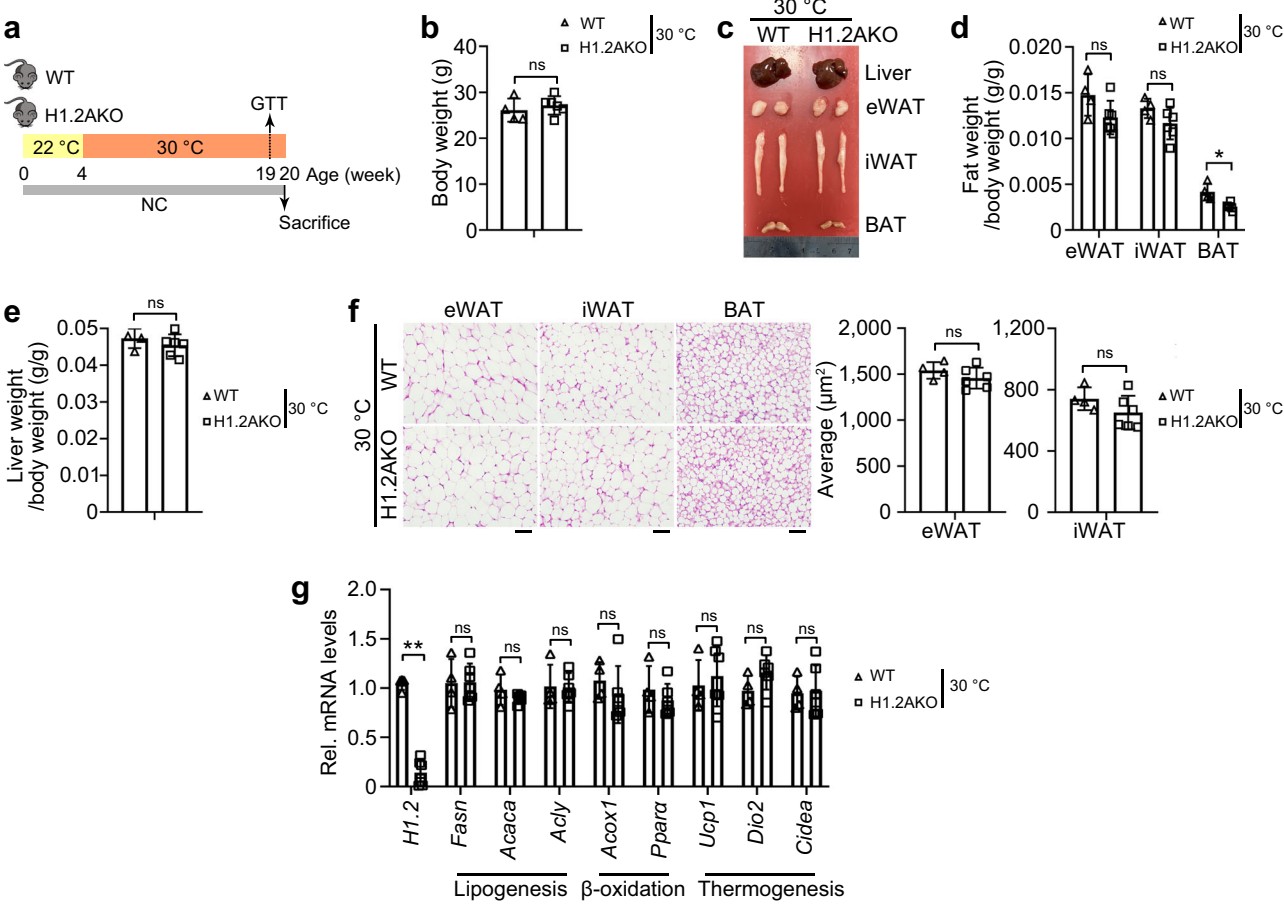

**Fig. 8 | Adipocyte H1.2 deficiency reduces weight gain through adaptive thermogenesis. a** Experimental design. Normal chow (NC)-fed WT mice and H1.2AKO mice were housed at 30 °C from 4-week-old for 15 weeks. **b–e** Body weight (**b**), anatomy (**c**) and different tissue weights (**d**, **f**) of NC-fed WT and H1.2AKO mice housed at 30 °C (WT mice, $n = 4$; H1.2AKO mice, $n = 6$; unpaired two-tailed Student's *t* test). **f** Representative H&E staining of eWAT/iWAT/BAT and average adipocyte area of eWAT and iWAT of WT and H1.2AKO mice housed at 30 °C (WT mice, $n = 4$; H1.2AKO mice, $n = 6$). Scale bar = 50 μm. **g** qPCR of genes related to lipogenesis, lipid oxidation and thermogenesis in iWAT of WT and H1.2AKO mice housed at 30 °C (WT mice, $n = 4$; H1.2AKO mice, $n = 6$; ns not significant; unpaired two-tailed Student's *t* test). Data are mean ± S.D. *$P < 0.05$, **$P < 0.01$; ns not significant. Source data and exact *P* values are provided in a Source data file.

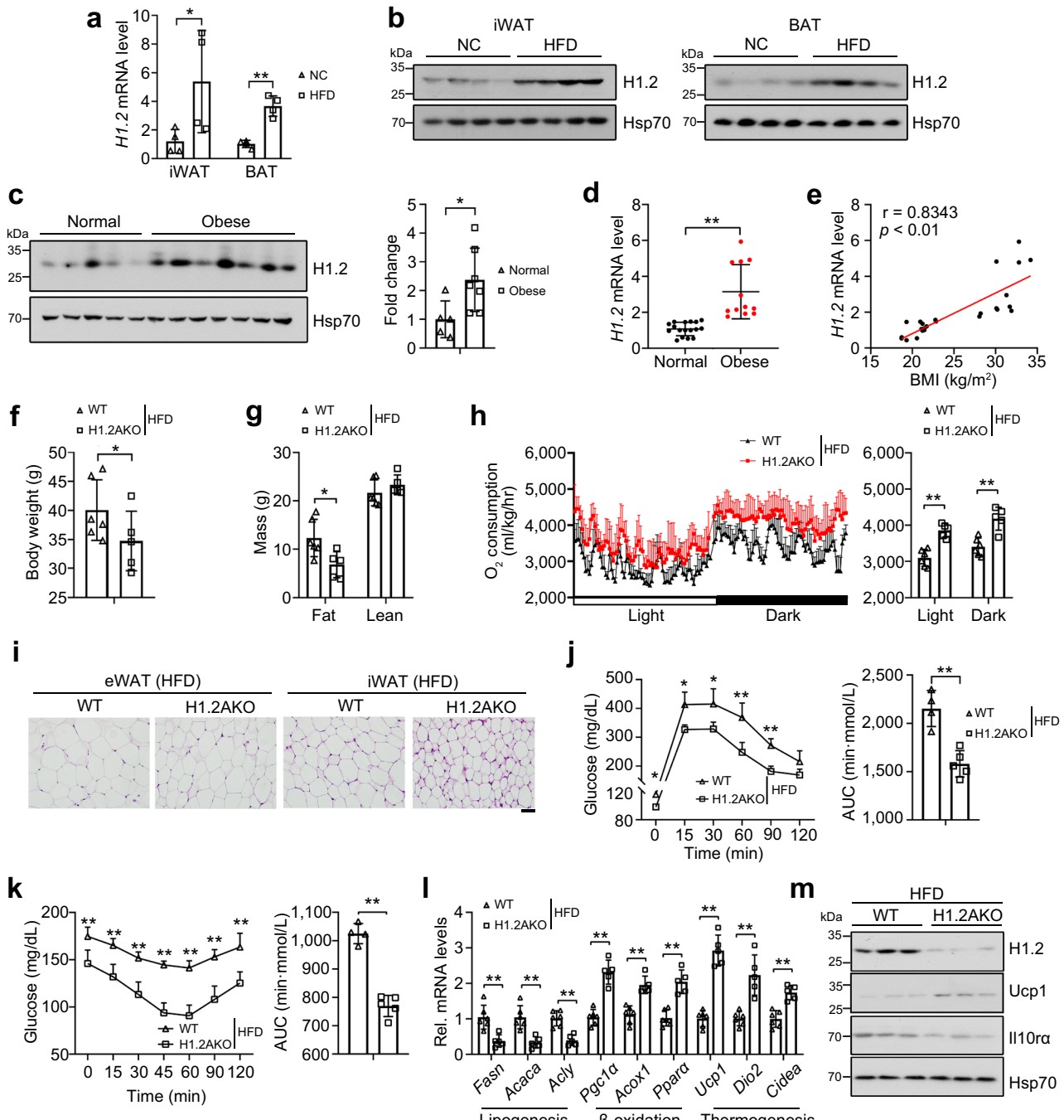

**Fig. 9 | H1.2AKO mice are resistant to HFD-induced obesity. a**, **b** mRNA (**a**) and protein (**b**) levels of H1.2 in iWAT and BAT of normal chow (NC)-fed or six-month HFD-fed WT mice (n = 4 per group; unpaired two-tailed Student's t test). **c** H1.2 level in human subcutaneous adipose tissues (n = 5 for normal and n = 7 for obese; unpaired two-tailed Student's t test). **d**, **e** qPCR of H1.2 in human subcutaneous adipose tissues (**d**) and correlation of H1.2 mRNA level with body mass index (BMI) (**e**) (n = 17 for normal and n = 12 for obese; unpaired two-tailed Student's t test). **f**, **g** body weight (**f**), fat mass and lean mass (**g**) of 21-week HFD-fed WT and H1.2AKO mice (WT mice, n = 6; H1.2AKO mice, n = 5; unpaired two-tailed Student's t test). **h** Oxygen consumption of HFD-fed WT and H1.2AKO mice (WT mice, n = 6;

H1.2AKO mice, n = 5; two-tailed ANCOVA with body weight as a covariate). **i** Representative H&E staining of eWAT/iWAT/BAT of HFD-fed WT and H1.2AKO mice (WT mice, n = 6; H1.2AKO mice, n = 5). Scale bar = 50 μm. **j**, **k** glucose tolerance test (**j**) and insulin tolerance test (**k**) of HFD-fed WT and H1.2AKO mice (WT mice, n = 4; H1.2AKO mice, n = 5; unpaired two-tailed Student's t test). **l** qPCR of genes associated with lipogenesis, lipid β-oxidation and thermogenesis in iWAT of HFD-fed WT and H1.2AKO mice (WT mice, n = 6; H1.2AKO mice, n = 5; unpaired two-tailed Student's t test). **m** H1.2, Ucp1, and Il10rα protein levels in iWAT of HFD-fed WT and H1.2AKO mice (n = 3 per group). Data are mean ± S.D. *P < 0.05, **P < 0.01. Source data and exact P values are provided in a Source data file.

## H1.2AKO mice protect against adiposity through Il10rα

Next, we assessed whether H1.2 inhibits fat accumulation Il10rα-dependently. NC-fed H1.2AKO mice were in situ injected with AAV-Il10rα in iWAT pads of both sides at 10-week-old, and kept for another 15 weeks under normal conditions (Fig. 10a, b). Il10rα overexpression

attenuated the effects of the adipocyte H1.2 deletion on reducing body weight, fat mass, and eWAT/iWAT/BAT weights under long-term NC-feeding (Fig. 10c–e). Downregulated blood glucose level, improved glucose tolerance, and insulin sensitivity in H1.2AKO mice were also abrogated upon Il10rα overexpression (Fig. 10f–h). Importantly, the

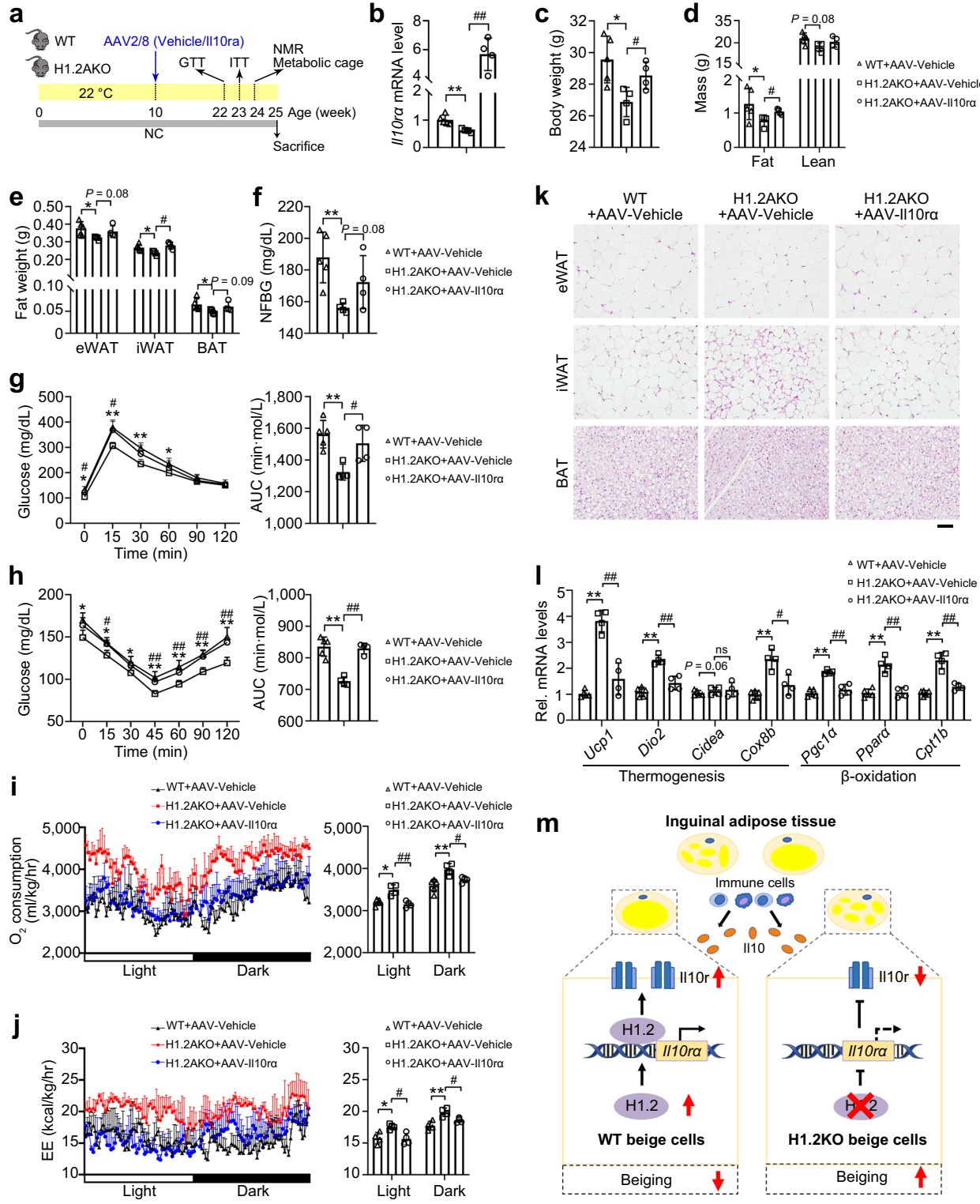

elevated $O_2$ consumption and energy expenditure, and the smaller adipocyte sizes in eWAT and iWAT, of H1.2AKO mice were also reversed by Il10rα overexpression (Fig. 10i–k). Consistently, qPCR showed the upregulations of thermogenic and lipid β-oxidation genes in H1.2AKO mice were also inhibited after Il10rα overexpression (Fig. 10l). Similarly, when overexpressing Il10rα in iWAT pads of both sides of HFD-fed WT and H1.2AKO mice, the differences observed in HFD-fed WT and H1.2AKO mice, including body weight, fat weight, glucose tolerance, and insulin sensitivity, as well as the thermogenic genes expression, were abolished (Supplementary Fig. 14).

Collectively, these data demonstrated that adipocyte H1.2 promotes adiposity and metabolic disorder via Il10rα.

## Discussion

Obesity, resulting from a chronic imbalance between energy intake and expenditure, is closely associated with cardiovascular diseases and type 2 diabetes[36]. Beige adipocytes possess thermogenic characteristics of brown adipocytes and contribute to whole-body energy expenditure[3], which makes them ideal targets to counteract obesity and associated metabolic disorders. Here, we provide evidence that

**Fig. 10 | Overexpression of Il10rα abolishes the metabolic improvement of H1.2AKO mice. a** Experimental design. Normal chow (NC)-fed WT and H1.2AKO mice were injected with AAV-Vehicle or AAV-Il10α at 10-week-old, and maintained on NC-fed conditions until 25-week-old. **b** *Il10rα* level in iWAT of AAV-Vehicle or AAV- Il10α injected NC-fed WT and H1.2AKO mice. **c**–**e** Body weight (**c**), fat mass (**d**), and tissue weights (**e**) of AAV-Vehicle or AAV-Il10α injected NC-fed WT and H1.2AKO mice. **f**–**h** Blood glucose (**f**), glucose tolerance test (**g**), insulin tolerance test (**h**) of AAV-Vehicle or AAV-Il10α injected NC-fed WT and H1.2AKO mice. **i, j** Oxygen consumption (**i**) and energy expenditure (EE) (**j**) of AAV-Vehicle or AAV-Il10α injected NC-fed WT and H1.2AKO mice. **k** Representative H&E staining of eWAT/iWAT of WT and H1.2AKO mice. Scale bar = 50 μm. **l** qPCR of genes related to thermogenesis and lipid β-oxidation in iWAT of AAV-Vehicle or AAV-Il10rα injected

NC-fed WT and H1.2AKO mice. For panels **a**–**l**, $n = 5$ for AAV-Vehicle injected WT mice, $n = 4$ for AAV-Vehicle or AAV-Il10rα injected H1.2AKO mice. Two-tailed ANCOVA with body weight as a covariate was used in **i** and **j**; otherwise, unpaired two-tailed Student's $t$ test was used for comparing two indicated groups. Data are mean ± S.D. *$P < 0.05$, **$P < 0.01$, AAV-Vehicle injected H1.2AKO mice compared with AAV-Vehicle injected WT mice; #$P < 0.05$, ##$P < 0.01$, AAV-Il10rα injected H1.2AKO mice compared with AAV-Vehicle injected H1.2AKO mice. Source data and exact $P$ values are provided in a Source data file. **m** Summary of our findings. In WT beige adipocytes, H1.2 binds to *Il10rα*, thus upregulating *Il10rα* transcription. Increased Il10rα reduces iWAT browning. While in H1.2 deletion beige adipocytes, reduced *Il10rα* level and increased browning are observed.

linker histone H1.2 is a negative modulator of browning in iWAT and affects thermogenesis in a beige adipocyte autonomous manner. In addition, we demonstrate a mechanistic link between H1.2 and Il10rα, showing that H1.2 activates Il10rα expression and manipulation of Il10rα level is sufficient to negate the anti-thermogenic effects of H1.2.

Some physiological functions of histone H1 variants have been reported in literature. H1.2/H1.3/H1.4 triple knockout mice are embryonic lethal, while single or double knockout of H1.2/H1.4 shows no developmental phenotype, suggesting that individual somatic H1 variant is dispensable for mouse development[20,37,38]. Whereas H1.2/H1.4 double knockout mice develop aggressive lymphomas[23], H1.2 whole-body knockout mice are resistant to DEN-induced hepatocarcinogenesis[24]. However, the role of a single histone variant in a specific tissue has not been reported in literature, herein, by studying adipocyte-specific H1.2 knockout mice, we revealed some effects of linker histone H1.2 on metabolic functions.

We demonstrated that linker histone 1.2 is enriched in mouse BAT and iWAT (Fig. 1a, b). Each histone H1 variant has a distinct expression profile in different cell types and tissues[39]. Here, abundant *H1.2* with extremely low levels of other somatic H1 variants were found in mouse iWAT and BAT (Supplementary Fig. 1a, b), implying a functional significance of H1.2 in adipose tissue. It is noteworthy that the level of H1.2, as an essential nucleosome component, is affected by environmental temperature (Fig. 1), which let us reason that cold-induced H1.2 may be a modulator of thermogenesis.

We also provided evidence that linker histone H1.2 is a brake of browning. Adipocyte-specific H1.2 deletion mice showed improved cold tolerance and enhanced iWAT browning; while mice in situ overexpressing H1.2 in iWAT showed attenuated cold tolerance and reduced iWAT browning (Fig. 3), these findings suggested the critical role of adipocyte H1.2 in regulating thermogenesis. Furthermore, higher energy expenditure in adipocyte-specific H1.2 knockout mice was associated with an improved metabolic phenotype, which disappeared when housed at 30 °C (Fig. 8), suggesting the metabolic effects of H1.2 were through its role in thermogenesis. Our results also suggest that H1.2 is an indicator of metabolic states, because both the mRNA and protein levels of H1.2 were increased in iWAT and BAT of HFD-fed mice, and upregulated H1.2 in subcutaneous WAT of obese human subjects was positively correlated with BMI (Fig. 9).

ChIP-seq results of cold-exposed iWAT showed that, compared with untreated iWAT, different H1.2 bindings were enriched in the promoter regions of genome (Supplementary Fig. 11). It has been reported that H1.2 may regulate gene transcription by affecting epigenetic marks on core histones[40]. We previously reported that H1.2 upregulates histone deacetylases, SIRT1 and HDAC1, to maintain the deacetylation status of H4K16, which increases transcriptions of autophagy-related genes that promotes autophagy in retinal cells[22]. Mass-spectrometry study also has reported distinct change in H3K27me3 and H3K36me3 levels after H1.2 knockout in human acute myeloid leukemia cells[23]. Furthermore, several epigenetic enzymes have been reported to bind with H1.2, including DNA methylase DNMT3a and histone deacetylase HDAC3, which play critical roles in

browning and thermogenesis[11,41]. Whether adipocyte H1.2 affects histone modifications and whether these modifications contribute to thermogenesis await further investigation.

Mechanistically, we found that loss of H1.2 in adipocytes downregulates Il10rα expression in iWAT (Fig. 4). By using adipocytes fractionated from iWAT and primary beige adipocyte, we found that H1.2 regulated Il10rα and thermogenesis in a beige cell autonomous manner (Figs. 4 and 5). H1.2 promoted *Il10rα* transcription via binding to its promoter, and this effect depends on the globular domain and C-terminal domain of H1.2 (Supplementary Fig. 10a–e). However, other factors that directly regulate Il10rα in adipocytes remain unclear. Published ChIP-seq data suggested PPARγ enrichment at the enhancer region of the *Il10rα* gene locus in adipocytes but without further experimental verification[42]. It has been reported that the transcriptional level of *l10rα* is upregulated in Bcl6 KO lymphoma cells[43], whether this regulation exists in adipocyte is unknown. Besides *Il10rα*, ChIP-seq suggested that there are more than 10000 genes bind with H1.2, and more than 1000 genes show altered H1.2 binding affinity after cold exposure (Supplementary Fig. 11). Whether these genes are direct targets of H1.2 and whether/how these contribute to browning need further investigation.

Interestingly, the effect of H1.2 on thermogenesis is observed in iWAT, but not BAT. In RNA-seq, 201 genes are commonly changed in iWAT and BAT of H1.2AKO mice, which only account for 27% or 20% of all altered genes in iWAT or BAT, respectively (Fig. 2d), indicating that H1.2 affects many different genes in these two fat pads. Humoral immune response was enriched by GO analysis in iWAT but not in BAT (Fig. 4a and Supplementary Fig. 8a). The different thermogenic effect on iWAT *vs* BAT probably because that H1.2 binds to different factors in beige and brown adipocytes, therefore exhibiting different gene regulatory patterns.

In this work, we report a function of H1.2 as a brake on beige adipocyte, and the effects of adipocyte H1.2-Il10rα axis on browning and metabolic states (Fig. 10m), which may involve in the pathogenesis of obesity related metabolic diseases and warrant future study.

## Methods
### Study approval
Human subcutaneous fat depots from both genders were obtained during surgery in Affiliated Tongji Hospital of Tongji Medical College (Wuhan, China), and approved by the ethics committee of Tongji Medical College in accordance with the principle of the Helsinki Declaration. Informed consent from all participants was obtained. Analysis of patient samples were not disaggregated reported for gender due to limited number of participants. Obese or normal subjects were defined as body mass index (BMI) $\geq 27$ or $18 \leq BMI \leq 23$, respectively[44]. Animals were handled according to the Guidelines of the China Animal Welfare Legislation, as approved by the Committee on Ethics in the Care and Use of Laboratory Animals of College of Life Sciences, Wuhan University.

## Animals

LoxP-flanked H1.2 (*H1.2^flox/flox^*) mice in C57BL/6 background, constructed by Saiye Inc. (Suzhou, China), and *adiponectin*-Cre mice (JAX, No.010803) were used to generate adipocyte-specific H1.2 knockout (H1.2AKO) mice. Genotyping was performed (primers listed in Supplementary Table 1), and male *H1.2^flox/flox^* mice with or without Cre recombinase identified as H1.2AKO or WT controls were used in this study; while female mice were used for breeding. C57BL/6 male mice were purchased from Hubei Provincial Center for Disease Control and Prevention. All mice were maintained in a specific-pathogen-free, temperature-controlled ($22 \pm 2\,°C$), humidity-controlled ($55 \pm 5\,\%$) animal facility with a 12-h light/dark cycle, and free access to water and normal chow (#1025; HFK Bio., Beijing, China), which was defined as normal conditions. The mice were euthanized by $CO_2$ inhalation.

## Diets and treatments

For cold-exposure, 10 weeks-old mice were single-housed at 6 °C for 3 days. Core body temperature was monitored using BAT-12 Microprobe Thermometer (Physitemp Inc., Clifton, NJ). For thermoneutral experiments, mice were housed at 30 °C for one month starting at 10-week-old, or for four months starting at 4-week-old. For $\beta_3$-adrenergic receptor agonist treatment, 10 weeks-old mice were intraperitoneally injected daily with CL316,243 (0.5 mg/kg body weight; Sigma, Saint Louis, MO), and sacrificed at day 1, 3, or 7 after first injection[31]. For diet-induced obesity, mice were fed with a HFD (60% Kcal fat, D12492; Research Diets, New Brunswick, NJ) at 4-week-old for 5–6 months. For adeno-associated virus (AAV) injections, $2.0 \times 10^{11}$ vg virus-packed mouse Il10rα (AAV-Il10rα), mouse H1.2 (AAV-H1.2) or control (AAV-Vehicle) (Obio Tech., Shanghai, China) was injected at multiple sites on the inguinal fat pads. Three weeks after injection, mice were challenged with 6 °C for 3 days; otherwise, the injected mice were maintained on NC or HFD for indicated time.

## Metabolic studies

Body composition was scanned by a mini-spec LF-50 analyzer (Bruker, Rheinstetten, Germany)[45]. For energy expenditure measurement, single-housed mice were acclimated for 24 h to the metabolic chamber of Comprehensive Laboratory Animal Monitoring System (CLAMS, Columbus Instruments, Columbus, OH) at room temperature. $O_2$ consumption and $CO_2$ production were monitored and data were collected by Oxymax (Columbus, OH). Energy expenditure-(EE) was calculated as $EE = (3.185 + 1.232 \times VO_2/VCO_2) \times VO_2$[46]. Mice were kept in metabolic chamber at 22 °C or 6 °C for 3 days[47].

## Glucose tolerance test and insulin tolerance test

For glucose tolerance test (GTT), mice were fasted for 16 hours, then injected intraperitoneally with D-glucose (1.5 g/kg body weight; Amresco, Solon, OH). Blood glucose was measured using a OneTouch blood glucose meter (Life Scan) at 0, 15, 30, 60, 90, and 120 min after injection[48]. For insulin tolerance test (ITT), mice were fasted for 6 hours and then injected intraperitoneally with insulin (0.75 unit/kg body weight; Lily, France). Blood glucose was measured at 0, 15, 30, 45, 60, 90, and 120 min after injection[48].

## Histological, immunohistochemical and immunofluorescent staining analysis

For adipocyte size analysis, tissues sections were stained with hematoxylin-eosin (H&E), and pictures of 4–10 different fields per sample were taken under an Olympus BX60 microscope equipped with a digital CCD. The area of adipocytes was manually traced and 400–1000 adipocytes for each sample were measured and analyzed using Image J software (Version 1.53)[47]. For immunohistochemical studies, sections of iWAT and BAT were incubated overnight with primary antibodies (information provided in Supplementary Table 2), then incubated with biotinylated goat anti-rabbit secondary antibody (BA-1000, 1:1000 dilution; Vector laboratories, Burlingame, CA). Positive staining was visualized using DAB substrate (Vector laboratories, Burlingame, CA). For immunofluorescent staining, primary antibodies (Supplementary Table 2) were applied overnight, and sections were then incubated with Alexa Fluor 488-conjugated rabbit anti-goat (A11078, 1:1000 dilution; Thermo Fisher Scientific, Waltham, MA) and/or Alexa Fluor 594-conjugated goat anti-rabbit (A11012, 1:1000 dilution; Thermo Fisher Scientific) secondary antibodies followed by DAPI staining[49]. Images were taken with a confocal microscopy (Leica SP8, Germany) or a HiS-SIM (High Sensitivity Structured Illumination Microscope, Computational Super-resolution Biotech, Guangzhou, China). For HiS-SIM, Z-stacks with a thickness of 4.1 µm (41 images) were collected.

## Cell culture and transfection

HEK293T cells (CL-0005, Procell Biotech, Wuhan, China) were cultured in DMEM media (Hyclone, South Logan, UT) plus 10% FBS (Lonsera, Shanghai, China). To knockdown H1.2, cells were transfected with pSuper vector containing scrambled shRNA or shRNAs targeting *H1.2* (primers provided in Supplementary Table 1). Flag-tagged H1.2, H1.2ΔN, H1.2ΔG-Flag, pRK-H1.2ΔN, H1.2A17V, H1.2K81N, H1.2G83A, H1.2S102F, H1.2P118S, and H1.2A171P were constructed into pRK vector with standard procedures, and transfected into cells.

## Isolation of mouse SVF and mature adipocytes

Minced iWAT and BAT were digested at 37 °C in digestion buffer containing collagenase D (Sigma) and Dispase II (Roche, Mannheim, Germany), then filtered through a 70 µm mesh, mature adipocytes, and SVF cells were separated by centrifugation[15]. Some SVF cells were cultured for in vitro differentiation experiments, while some SVF cells and mature adipocytes were collected for further analysis.

## Primary adipocyte differentiation, infection, treatment, and Oil Red O staining

Mouse SVF cells were grown in DMEM/F12 media (Hyclone) supplemented with 10% FBS. For beige or brown adipocyte differentiation, confluent cells (D0) were changed to medium A containing 0.5 mM IBMX, 1 µM dexamethasone, 125 nM indomethacin, 850 nM insulin, 1 nM T3 and 1 µM rosiglitazone (all from Sigma, Saint Louis, MO) for 2 days, then changed to medium B containing 850 nM insulin, 1 nM T3 and 1 µM rosiglitazone for another 5 days. H1.2 knockout or Il10rα overexpression was achieved by infecting differentiated SVF cells which were isolated from *H1.2^flox/flox^* mice with adenovirus-packed Cre-GFP or AAV packed Il10rα (Obio Tech.; MOI = 100) at D2. At D7, differentiated beige adipocytes were incubated with or without 10 µM isoproterenol (Sigma) for 4 h. For Il10 stimulation, 50 ng/ml or 100 ng/ml recombinant mouse Il10 (Peprotech, Rocky Hill, NJ) was added to differentiated beige adipocytes for 12 h. For Oil Red O staining, differentiated cells were fixed with 4% formaldehyde, and stained with 0.5% Oil Red O (Sigma) dissolved in propylene glycol at D7[15,17,31].

## Quantitative real-time PCR

Total RNA was extracted using RNAiso Plus (Takara Biotech., Dalian, China). Equal amount of RNA of each sample was reverse transcribed into cDNA using M-MLV (Invitrogen, Grand Island, NE)[50]. Quantitative real-time PCR (qPCR) was performed using target-specific primers (Supplementary Table 1) and SYBR Green probes (Yeasen Biotech., Shanghai, China). Relative transcripts were calculated using the comparative $C_T$ method and *Rn18s* was used as the internal control[50].

## RNA-sequencing

Total RNA of iWAT and BAT of WT and H1.2AKO mice was isolated individually. Equal amount of RNA ($n = 4$) from the same group was

combined into 2 samples/group for RNA-seq as previously reported[51]. Sequencing and data analysis were performed by Novogene Bioinformatics (Beijing, China) as previously reported[52].

## Chromatin immunoprecipitation sequencing and ChIP-qPCR analysis

iWAT tissues were cut and fixed with 1% formaldehyde and stopped by adding glycine. Pellets were homogenized and resuspended in RIPA buffer (Beyotime Biotech.), and crosslinked chromatin was sheared into 200-1000 bp (for ChIP-qPCR) or 200-500 bp fragments (for ChIP-seq) with a sonicator (Scientz Biotech., Ningbo, China) or Bioruptor Sonication System UCD-300 (Diagenode, Denville, NJ), respectively. 200 μg DNA from each sample was resuspended in RIPA buffer containing 2 μg H1.2 antibody (ab181973; Abcam, Cambridge, UK) plus protein G magnetic beads (Bio-Rad, Hercules, CA) for overnight. Immune complexes were collected, washed, and eluted. The DNA-protein crosslinks were then incubated overnight at 65 °C for extracting DNA for qPCR analysis or high through-put sequencing. For qPCR analysis, 4 different regions of *Il10rα*, or *Nrp2*, or *Gapdh* promoter (Supplementary Table 1) were used, with the input samples used as internal controls. High through-put sequencing and analysis were conducted by Seqhealth Tech. (Wuhan, China) using DNBSEQ-T7 sequencer (MGI Tech., China) with PE150 model. Raw data were filtered by Trimmomatic (Version 0.36). The clean reads were mapped to mouse genome (GRCm38) using STAR software (Version 2.5.3a). The RSeQC (Version 2.6) was used for reads distribution analysis. The MACS2 software (Version 2.1.1) was used for peak calling. The bedtools (Version 2.25.0) was used for peaks annotation and peak distribution analysis. The differentially binding peaks were identified by a python script, using fisher test. Visualization of mapping results was achieved by IGV (Version 2.16.1).

## Western Blots

Cells or tissues were harvested and sonicated in RIPA buffer (Beyotime Biotech., Shanghai, China) supplemented with protease inhibitor cocktail (Roche). 20–50 μg protein per sample was separated by SDS-PAGE and electroblotted onto PVDF membrane (Millipore, Billerica, MA). Primary antibodies (Supplementary Table 2) were incubated overnight, then blots were incubated with HRP-conjugated goat anti-rabbit (1706515, 1:5000 dilution; Bio-Rad) or HRP-conjugated goat anti-mouse (1706516, 1:10000 dilution; Bio-Rad) secondary antibodies. Targeted protein bands were visualized by enhanced chemiluminescence reagent (Beyotime Biotech.) and evaluated using Quantity One (Version 4.6.2)[53,54].

## Luciferase reporter assay

Luciferase reporter assays were performed using a dual-specific luciferase assay kit (Promega, Madison, WI) as previously reported[24,55]. *Il10rα* promoter from −2000 to TSS (transcription start site) was cloned into pGL3-basic firefly luciferase reporter vector. HEK293T cells were transiently co-transfected with the reporter plasmid, Renilla luciferase plasmid, and indicated plasmid. Cells were lysed 24 hours after transfection, firefly luciferase activity were determined and normalized by Renilla luciferase activity.

## Statistics and reproducibility

All the results were expressed as the mean ± S.D. (standard deviation). The exact sample size (*n*) for each experimental group was indicated in figure legends. For correlation analysis, linear regression was performed. All statistical analyses were performed using GraphPad Prism 8 (La Jolla, CA). For comparisons between two unpaired independent groups, two-tailed Student's *t* test was used; for comparisons of multiple groups, one-way ANOVA (analysis of variance) with Tukey's or Dunnet's test was used. Two-tailed ANCOVA (analysis of covariance) was used to compare the metabolic studies with body weight as a covariate. $P < 0.05$ was considered statistically significant. * and ** correspond to $P$ values of <0.05 and <0.01, respectively.

## Reporting summary

Further information on research design is available in the Nature Portfolio Reporting Summary linked to this article.

## Data availability

All data relating to this study can be found in the main text, figures, or supplementary information. RNA-seq data described in this study are available in NCBI GEO database under accession code GSE215412. ChIP-seq data generated in this study are accessible in NCBI GEO database through accession code GSE232530. Source data are provided with this paper.

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

## Acknowledgements

We thank Dr. Zhuzhen Zhang from Wuhan University for helpful discussions. This work is supported by the National Key R&D Program of China (2018YFA0800700 and 2019YFA0802701 to L.Z.), the Natural Science Foundation of China (32021003 to L.Z.; 82273838 to K.H.), the Natural Science Foundation of Hubei Province (2021CFA004 to K.H. and 2021CFB250 to C.L.), the Wuhan Science and Technology Bureau Innovation Project (2022020801020526 to C.L.), and the Fundamental Research Funds for the Central Universities (2042022dx0003 to L.Z.).

## Author contributions

L.Z. conceived the project. Y.Y. and L.Z. designed the experiments. Y.Y., Y.F., Y.Z., R.Q., W.K., YuL., J.S., and C.L. performed the experiments and analyzed the data. Y.C. assisted with the figure drawing. C.W. and M.W. assisted with the ChIP-seq data analysis. YK L provided the human samples. Y.Y., L.Z., YongL. and K.H. wrote the manuscript.

## Competing interests

The authors declare no competing interests.
