## [Peer Review File · Nature Communications]

Linker histone variant H1.2 is a brake on white adipose tissue
browningREVIEWER COMMENTS

Reviewer #1 (Remarks to the Author):

In this very extensive manuscript, the authors present 3 key findings; First they establish that linker Histone 1.2 is enriched in iWAT and BAT and is positively regulated by cold or b-adrenergic stimuli; they then continue to demonstrate very convincingly that H1.2 is a negative regulator of browning in iWAT. And thirdly, they draw a mechanistic link between H1.2 and IL10ra, showing that H1.2 activates IL10ra expression and that manipulation of IL10ra levels downstream of H1.2 are sufficient to rescue or negate the anti-thermogenic effects of H1.2.

As expected, a higher energy expenditure in mature adipocyte-specific H1.2 knock-out mice resulted also in an improved metabolic phenotype (lower body weight, better glucose and insulin sensitivity in old chow mice as well as HFD-fed mice).

Overall, the manuscript is very well written, the data is presented in a convincing and rational way and an ample amount of complimentary experiments has been conducted to corroborate the findings in different in-vivo and in-vitro settings.

Nevertheless, there are a few points that require further attention:

- 1) The expression levels of the other H1 variants in H1.2KO and overexpression settings should be provided – just to exclude that their levels are disproportionately affected by H1.2 absence/presence and the observed phenotype is also due to their dysregulation rather than only H1.2 itself.
- 2) I'm not convinced that the anti-obesogenic effect of H1.2 is significant enough to merit its emphasis in the title. Generally, the effect size on body weight is relatively small – especially when considering the number of mice in the respective experiments, which, with 4-5 animals per group is at the very low end of what is commonly used in comparable studies. Also, only a single time point is presented, which does not capture the whole story. I would therefore also recommend to remove the statement about the H1.2-IL10ra axis as a potential axis for obesity treatment. Nevertheless, the accompanying metabolic and molecular analyses focusing on the target tissue (iWAT) in these cohorts are very convincing in a sense that energy expenditure, glucose and insulin sensitivity, tissue morphology, as well as the expression of marker genes clearly support the presented hypothesis of H1.2 impeding iWAT browning.
- 3) Considering the nature of the protein of interest, I would have preferred to see a proper genomic analysis (ChIP-Seq, Cut and Tag, or similar) – comparing H1.2 chromatin distribution in iWAT and BAT, both during thermoneutral/RT and cold exposure conditions and to compare it with the matching gene expression data sets, as this would have helped to understand the uniqueness of H1.2 role in iWAT and aided in the mechanistic understanding of how H1.2 impairs a browning phenotype. Also, in this context, the ChIP-qPCR and Luciferase assays in Fig. 5j-n should be moved to the supplementary section. Fig5. j adds no real value as the ChIP results only reflect the expression levels of H1.2 under the respective conditions – (less in H1.2KO, more in cold condition). It represents no unbiased approach identifying H1.2 binding sites. Similarly, the luciferase assay, is more a study of H1.2 protein function and not so much proper evidence for its regulation of IL10ra transcription. In both cases the necessary controls (i.e. different genome/gene promoter regions) are missing.
- 4) I fail to see the point of Ext. Fig 9 and the associated text section – the enrichment of H1.2 was already presented in Ext. Fig 1c, upregulation of H1.2 in subcutaneous adipose tissue of obese individuals is convincingly shown in Fig.8
- 5) In general, the discussion of clinical significance of H1.2, variants etc. seems a bit premature and forced at this stage. The manuscript (discussion) would benefit from a focus on the novel findings (role of H1.2 in thermogenesis, H1.2 in iWAT vs BAT, H1.2-IL10ra axis, H1.2 molecular mode of action) for which the authors present sufficient and convincing data – without the need of forcing the clinical relevance at this stage.
- 6) Figure legend 5A mentions AAV-Cre, yet in the text (line 204) it says lentivirus-Cre

Reviewer #2 (Remarks to the Author):

The present study provided data to demonstrate that linker histone variant H1.2 inhibit beiging and thermogenesis of white adipose tissue and promote obesity. The authors identified Il10ra as the target of H1.2 and indicated H1.2 inhibited thermogenesis by upregulating transcription of Il10ra. The function of H1.2 in adipose metabolism is of novelty and interest. Yet, several points should be clarified before the study can be considered for publication.

Major issues

Levels of H1.2 in both WAT and BAT were induced by cold, and ablation of H1.2 in adipocyte also stimulated thermogenesis and energy expenditure. It made sense if the upregulation of H1.2 was the feedback to act as a braker for beiging process upon cold challenge. From this point of view, however, what would the phenotype be if H1.2 was overexpressed under cold scenario? Therefore, analysis of WAT with H1.2 gain of function is important to interperate the role of H1.2 in adipose metabolism.

H1.2 knockout had no thermogenic effect on brown adipocyte. Did the author have any proposal. Was the differentially expressed genes in Fig. 4a involved in BAT? Was H1.2-Il10ra involved in BAT?

Minor issues

To clarify beiging of adipocytes was through Il10ra on adipocyte, the level of Il10ra should be examined after fractionating cells into mature adipocytes and SVF in figure 4f&g, considering it was hard to determine whether Il10ra was on adipocytes in Fig.4e.

In fig.5, whether upregulation of Il10ra in H1.2 knockout adipocytes reverse the effect of il10?

In fig.10. the group of overexpression of Il10ra was missing. Is there any more targets than Il10ra by H1.2 in adipocytes? Or any more factors than H1.2 that can regulate Il10ra in adipocytes? Would the authors discuss?

In fig.5c-d, how about the expressions of PPAR α and relative genes for lipid metabolism?

The data were redundant and could be reorganized for concision. For example, fig.1, 2, 7, & 9 can be organized for solving one question, the same as Fig. 5, 6 &10.

In fig. S2b, the quality of western blot was poor, it's hard to say H1.2 was elevated by CL at protein level. Would the authors repeat the experiments?

Point-by-Point Response Letter

We would like to thank the reviewers for the insightful comments and suggestions, which have greatly helped to improve our study. Based on these comments, we have performed additional experiments, re-organized some of the data, and provided more relevant discussion in the revised manuscript. The revised places were shown in blue fonts for convenience, and please note that some figures have been renumbered as a result of inclusion of new data. Below are our point-by-point responses.

Reviewer #1:

In this very extensive manuscript, the authors present 3 key findings; First they establish that linker Histone 1.2 is enriched in iWAT and BAT and is positively regulated by cold or b-adrenergic stimuli; they then continue to demonstrate very convincingly that H1.2 is a negative regulator of browning in iWAT. And thirdly, they draw a mechanistic link between H1.2 and IL10ra, showing that H1.2 activates IL10ra expression and that manipulation of IL10ra levels downstream of H1.2 are sufficient to rescue or negate the anti-thermogenic effects of H1.2. As expected, a higher energy expenditure in mature adipocyte-specific H1.2 knock-out mice resulted also in an improved metabolic phenotype (lower body weight, better glucose and insulin sensitivity in old chow mice as well as HFD-fed mice).

Overall, the manuscript is very well written, the data is presented in a convincing and rational way and an ample amount of complimentary experiments has been conducted to corroborate the findings in different in-vivo and in-vitro settings.

Reply: We thank the reviewer for the encouraging comments that “the manuscript is very well written, the data is presented in a convincing and rational way”. In the revised manuscript, we have carefully addressed the reviewer’s comments with details listed below.

Nevertheless, there are a few points that require further attention:

1) The expression levels of the other H1 variants in H1.2KO and overexpression settings should be provided – just to exclude that their levels are disproportionately affected by H1.2 absence/presence and the observed phenotype is also due to their dysregulation rather than only H1.2 itself.

Reply: We appreciate this great point. Following the reviewer's suggestions, we performed qPCR to examine the transcriptional levels of the other somatic H1 variants in H1.2AKO mice under both normal and cold conditions. Compared to WT mouse, no significant change of other somatic H1 variants was found in iWAT or BAT of H1.2AKO mouse under normal conditions or upon cold exposure. These pieces of new data were provided in revised Supplementary Fig. 3h-i, Supplementary Fig. 5a and Supplementary Fig. 6d, respectively.

Furthermore, gain-of-function of H1.2 in iWAT was achieved by *in situ* injection of AAV2/8-H1.2 in mice, and no significant change in transcriptional levels of the other somatic H1 variants was detected (revised Supplementary Fig. 7d).

2) I'm not convinced that the anti-obesogenic effect of H1.2 is significant enough to merit its emphasis in the title. Generally, the effect size on body weight is relatively small – especially when considering the number of mice in the respective experiments, which, with 4-5 animals per group is at the very low end of what is commonly used in comparable studies. Also, only a single time point is presented, which does not capture the whole story. I would therefore also recommend to remove the statement about the H1.2-IL10ra axis as a potential axis for obesity treatment. Nevertheless, the accompanying metabolic and molecular analyses focusing on the target tissue (iWAT) in these cohorts are very convincing in a sense that energy expenditure, glucose and insulin sensitivity, tissue morphology, as well as the expression of marker genes clearly support the presented hypothesis of H1.2 impeding iWAT browning.

Reply: We thank the reviewer for this thoughtful suggestion. Accordingly, in the revised manuscript, the title has been revised as “Linker histone variant H1.2 is a

brake on white adipose tissue browning”. Also, the statement about the H1.2-IL10ra axis as a potential pathway for obesity treatment has been removed.

3) Considering the nature of the protein of interest, I would have preferred to see a proper genomic analysis (ChIP-Seq, Cut and Tag, or similar) – comparing H1.2 chromatin distribution in iWAT and BAT, both during thermoneutral/RT and cold exposure conditions and to compare it with the matching gene expression data sets, as this would have helped to understand the uniqueness of H1.2 role in iWAT and aided in the mechanistic understanding of how H1.2 impairs a browning phenotype.

Also, in this context, the ChIP-qPCR and Luciferase assays in Fig. 5j-n should be moved to the supplementary section. Fig. 5j adds no real value as the ChIP results only reflect the expression levels of H1.2 under the respective conditions – (less in H1.2KO, more in cold condition). It represents no unbiased approach identifying H1.2 binding sites. Similarly, the luciferase assay, is more a study of H1.2 protein function and not so much proper evidence for its regulation of IL10ra transcription. In both cases the necessary controls (i.e. different genome/gene promoter regions) are missing.

Reply: We agree with the reviewer that genomic analysis of H1.2 chromatin distribution will help to understand the mechanistic roles of H1.2 in adipose tissues. Following this suggestion, we first tried Cut & Tag assay. This assay requires dispersing the fresh iWAT tissue into single cell suspension, then enriching adipocytes with ConA-coated beads. Due to the large amount of lipid in dispersed adipose tissues which wrapped around and clumped the beads, we tried several times and were unable to enrich adipocytes with this approach. To our knowledge, no Cut & Tag assay for adipose tissues has been reported in literature. After consulting several technical supports, we next performed ChIP-seq of H1.2 under room temperature or cold exposure conditions in iWAT. BAT was not used because of limited amount of the antibody and tissues. Overall, 10418 and 12547 genes were detected in iWAT DNA immunoprecipitated by H1.2 antibody under room temperature or cold exposure, respectively. Among them, more than 50% of H1.2 binding sites were distributed in

intergenic regions; 13-15% in promoter plus transcriptional start site regions (revised Supplementary Fig. 11a), which is consistent with the ChIP-seq results targeting histone H3.3 or histone modification such as H3K9me3 (Emeline Fontaine et.al, Nucleic Acids Res, 2022; Zhen Wang et.al, Adv Sci, 2021). Cold stimulation induced H1.2 enrichment on genomes, especially on promoter regions (revised Supplementary Fig. 11a-b). Gene ontology analysis of ChIP-seq results revealed a significant enrichment in biological processes including lipid metabolism, oxidative phosphorylation and immune response after cold stimulation, which further implies the thermogenic function of H1.2 in iWAT (Supplementary Fig. 11c).

Notably, stronger binding of H1.2 in *Il10ra* gene upon cold exposure was observed based on the ChIP-seq analysis (revised Supplementary Fig. 11d), which was consistent with the ChIP-qPCR results.

Following the reviewer's suggestion, the ChIP-qPCR and luciferase assays have been moved to the revised Supplementary Fig. 10. To obtain more convincing ChIP-qPCR data, promoter regions of *Gapdh*, which showed no H1.2 binding, and promoter regions of *Nrp2* (recombinant neuropilin 2), which showed strong H1.2 binding but was not affected by cold exposure, were used respectively as negative controls (Supplementary Fig. 10a-b).

4) I fail to see the point of Ext. Fig 9 and the associated text section – the enrichment of H1.2 was already presented in Ext. Fig 1c, upregulation of H1.2 in subcutaneous adipose tissue of obese individuals is convincingly shown in Fig.8

Reply: We agree. The previous Supplementary Fig. 9 has been removed from the revised manuscript.

5) In general, the discussion of clinical significance of H1.2, variants etc. seems a bit premature and forced at this stage. The manuscript (discussion) would benefit from a focus on the novel findings (role of H1.2 in thermogenesis, H1.2 in iWAT vs BAT, H1.2-IL10ra axis, H1.2 molecular mode of action) for which the authors present

sufficient and convincing data – without the need of forcing the clinical relevance at this stage.

Reply: As the reviewer suggested, in the revised discussion we focused on the novel findings of our studies, and revised the part of clinical significance of H1.2 and relative mutations.

6) Figure legend 5A mentions AAV-Cre, yet in the text (line 204) it says lentivirus-Cre

Reply: We apologize for these typos. We used adenovirus packed Cre recombinases to infect the differentiated SVF cells isolated from H1.2^{fl^{ox}/fl^{ox}} mice. We have corrected these typos in the relevant results, methods and figure legends.

Reviewer #2:

The present study provided data to demonstrate that linker histone variant H1.2 inhibit beiging and thermogenesis of white adipose tissue and promote obesity. The authors identified Il10ra as the target of H1.2 and indicated H1.2 inhibited thermogenesis by upregulating transcription of Il10ra. The function of H1.2 in adipose metabolism is of novelty and interest. Yet, several points should be clarified before the study can be considered for publication.

Reply: We thank the reviewer for the encouraging comments that “the manuscript is of novelty and interest”. In the revised manuscript, we have carefully addressed the reviewer’s comments with details listed below.

Major issues

Levels of H1.2 in both WAT and BAT were induced by cold, and ablation of H1.2 in adipocyte also stimulated thermogenesis and energy expenditure. It made sense if the upregulation of H1.2 was the feedback to act as a braker for beiging process upon cold challenge. From this point of view, however, what would the phenotype be if H1.2 was overexpressed under cold scenario? Therefore, analysis of WAT with H1.2 gain of function is important to intemperate the role of H1.2 in adipose metabolism.

Reply: We thank the reviewer for the insightful comments. As the reviewer suggested, H1.2 was overexpressed by *in situ* injection of AAV2/8-H1.2 into iWAT pads of both sides in WT mice and kept for another 3 weeks before cold exposure. Under cold conditions, H1.2-overexpressing mice showed less cold tolerance and decreased iWAT browning (revised Fig. 3f-k), which is consistent with the opposite phenotypes of H1.2AKO mice under cold stress.

H1.2 knockout had no thermogenic effect on brown adipocyte. Did the author have any proposal. Was the differentially expressed genes in Fig. 4a involved in BAT? Was H1.2-Il10ra involved in BAT?

Reply: We appreciate this insightful point. The top ten biological processes enriched by GO pathway analysis in iWAT were different from those enriched in BAT. The new GO pathway analysis in BAT was provided in revised Supplementary Fig. 8a. As the reviewer suggested, we further examined the genes involved in immune response in BAT, which were significantly altered in iWAT of H1.2AKO mice. No changes were found in BAT, including B cell surface marker *Cd22* and *Cd45*, as well as in T cell surface marker *Cd3*, and chemokines associated with these cells. This piece of new data was provided in revised Supplementary Fig.8b. Previously, we showed similar mRNA levels of *Il10ra* in BAT (whole cell lysate) of WT and H1.2AKO mice under room temperature or cold conditions (Supplementary Fig. 8c-f), suggesting that the H1.2-Il10ra axis was not implicated in BAT.

Moreover, we re-checked our RNA-seq data shown in Fig. 2d. When comparing the differential expressed genes in iWAT and BAT in H1.2AKO mice, 201 genes were commonly changed, which only accounts for 27% of all the altered genes in iWAT and 20% of all the altered genes in BAT, indicating that H1.2 affects many different genes in these two fat pads. Thus, the unique thermogenic effect on iWAT rather than BAT probably indicates that H1.2 binds to different factors in beige and brown adipocytes, therefore exhibiting different gene regulatory patterns. We have discussed this in revised manuscript (Page 22-23).

Minor issues

To clarify being of adipocytes was through Il10ra on adipocyte, the level of Il10ra should be examined after fractionating cells into mature adipocytes and SVF in figure 4f&g, considering it was hard to determine whether Il10ra was on adipocytes in Fig.4e.

Reply: We appreciate this great point. As suggested, iWAT was fractionated into mature adipocytes and SVFs to examine the Il10ra level. Under normal condition, the expression of Il10ra was enriched in mature adipocytes rather than SVFs. Furthermore, decreased Il10ra level in mature adipocytes, but not in SVFs, was found in H1.2AKO mouse, which is consistent with our previous conclusion (revised Fig. 4g-h). Similar results were also obtained when fractionating iWAT into mature adipocytes and SVFs after three days of cold challenge. These pieces of new data were provided in the revised Fig. 4k-l.

In addition, the immunofluorescent staining of Il10ra in iWAT was re-performed to better present Il10ra location on adipocytes (revised Fig. 4e). We also provided an enlarged image to show the co-localization of Il10ra and Perilipin 1 (adipocyte marker) more clearly (revised Fig. 4e). Furthermore, using a HiS-SIM (High Sensitivity Structured Illumination Microscope), the XZ and YZ plots both demonstrated co-localization of Il10ra with Perilipin 1 on the cell membrane of adipocytes (revised Fig. 4f).

In fig.5, whether upregulation of Il10ra in H1.2 knockout adipocytes reverse the effect of il10?

Reply: We appreciate this insightful point. We overexpressed Il10ra vs vehicle in H1.2 knockout beige adipocytes during differentiation, and then treated mature beige adipocytes with IL10. In H1.2-knockout beige adipocytes, IL10 had no inhibitory effect on the expression of thermogenic related genes (*Ucp1* and *Dio2*). However, in Il10ra-overexpressing H1.2-knockout beige adipocytes, IL10 significantly reduced the

expression of *Ucp1* and *Dio2* (revised Fig. 5k-m). Thus, upregulation of *Il10ra* in H1.2-knockout beige adipocytes could reverse the effect of IL10.

In fig.10. the group of overexpression of *Il10ra* was missing. Is there any more targets than *Il10ra* by H1.2 in adipocytes? Or any more factors than H1.2 that can regulate *Il10ra* in adipocytes? Would the authors discuss?

Reply: We thank the reviewer for pointing this out. We agree that it is better to have another control group with overexpression of *Il10ra* in WT mice. Due to the limited amount of adeno-associated virus for *Il10ra* (AAV-*Il10ra*) available for the previous 9-month animal study, and because the most crucial question we would like to address was whether rescuing *Il10ra* expression in H1.2AKO mice could abolish the browning effects, we chose to focus on the effects of AAV-*Il10ra* treatment in H1.2AKO mice.

Il10ra may not be the only target of H1.2 in adipocytes. In our newly performed ChIP-seq analysis in iWAT, H1.2 was found to associate with more than 10000 genes, and more than 1000 genes were shown to have altered binding affinity with H1.2 after cold exposure, including *Il10ra*. However, whether these genes are direct targets of H1.2 and contribute to the anti-thermogenesis effect of H1.2 need further detailed investigations. To our knowledge, factors besides H1.2 that can directly regulate *Il10ra* in adipocytes remain unclear. Analysis of published ChIP-seq data revealed enrichment of PPAR γ at the enhancer region of the *Il10ra* gene locus in adipocytes (Siersbæk et al., Mol. Cell. Biol., 2012), but whether PPAR γ could directly regulate *Il10ra* transcription remains to be dissected. In addition, the transcriptional level of *Il10ra* was upregulated in Bcl6 KO lymphoma cell line (Beck D, J Biol Chem., 2016), but whether this regulation also exists in adipocyte is unknown. We discussed the possible target genes of H1.2 and the possible regulators for *Il10ra* in the revised manuscript (Page 21).

In fig.5c-d, how about the expressions of PPAR α and relative genes for lipid metabolism?

Reply: As the reviewer suggested, we examined the expressions of *PPAR α* and relative genes for lipid metabolism such as *Acox1*, *Acs11*, *Cpt1a* and *Cpt1b* in primary H1.2 knockout beige adipocytes. The mRNA levels of *PPAR α* , *Acox1*, *Acs11* and *Cpt1b* were significantly upregulated (revised Fig. 5e).

The data were redundant and could be reorganized for concision. For example, fig.1, 2, 7, & 9 can be organized for solving one question, the same as Fig. 5, 6 &10.

Reply: We thank the reviewer for this suggestion. After incorporating new data, we have re-organized some of the figures as suggested to present the data more concisely and logically. As the reviewer suggested, we moved the previous Fig. 9 in front of the previous Fig. 8. In the revised manuscript, we first established that H1.2 was positively regulated by cold and acted a negative regulator of browning in iWAT. Then we drew a mechanistic link between H1.2 and *Il10ra* in iWAT and beige adipocytes. Finally, we showed the metabolic role of H1.2 in long-term NC- and HFD-fed conditions, and these effects were dependent on *Il10ra* and thermogenesis.

In fig. S2b, the quality of western blot was poor, it's hard to say H1.2 was elevated by CL at protein level. Would the authors repeat the experiments?

Reply: As the reviewer suggested, we repeated the experiments and showed the new results with better quality in the revised Supplementary Fig. 2b.

REVIEWERS' COMMENTS

Reviewer #1 (Remarks to the Author):

I'd like to thank the authors for addressing the reviewers' comments in great detail and for performing the additional experiments that have been suggested. This definitely strengthens the manuscript, allowing them to draw a convincing picture of an iWAT axis involving H1.2, IL10ra and IL10 and the attenuating effect of H1.2 on iWAT browning. Similarly, the connected metabolic benefits of knocking out adipose H1.2 have been convincingly proven by a coherent series of overexpression/rescue and pharmacological studies. Mechanistically, I am also satisfied for the scope of this manuscript. I have no further questions and would like to congratulate the authors to their work.

Reviewer #2 (Remarks to the Author):

The questions have been addressed with new data, and the manuscript has been improved. The conclusion and discussion appear rational.

Point-by-Point Response Letter

We are grateful to both reviewers who invested time and effort in reviewing this manuscript. Their comments and suggestions are greatly helpful.

Reviewer #1:

I'd like to thank the authors for addressing the reviewers' comments in great detail and for performing the additional experiments that have been suggested. This definitely strengthens the manuscript, allowing them to draw a convincing picture of an iWAT axis involving H1.2, IL10ra and IL10 and the attenuating effect of H1.2 on iWAT browning. Similarly, the connected metabolic benefits of knocking out adipose H1.2 have been convincingly proven by a coherent series of overexpression/rescue and pharmacological studies. Mechanistically, I am also satisfied for the scope of this manuscript. I have no further questions and would like to congratulate the authors to their work.

Reply: We are pleased to learn that the revised manuscript addressed the reviewer's concerns and we thank the reviewer for the encouraging comments.

Reviewer #2:

The questions have been addressed with new data, and the manuscript has been improved. The conclusion and discussion appear rational.

Reply: We are happy to learn that our revised manuscript addressed the comments of the reviewer.